# Stacking Your Transformers: A Closer Look at Model Growth for Efficient LLM Pre-Training

**Wenyu Du**[1*]  **Tongxu Luo**[2,3*†]  **Zihan Qiu**[4]  **Zeyu Huang**[5]  **Yikang Shen**[6]
**Reynold Cheng**[1]  **Yike Guo**[2]  **Jie Fu**[2‡]

[1]School of Computing and Data Science, The University of Hong Kong    [2]HKUST
[3]USTB    [4]Tsinghua University    [5]University of Edinburgh    [6]MIT-IBM Watson AI Lab
wydu@cs.hku.hk    tongxuluo@gmail.com    jiefu@ust.hk

## Abstract

LLMs are computationally expensive to pre-train due to their large scale. Model growth emerges as a promising approach by leveraging smaller models to accelerate the training of larger ones. However, the viability of these model growth methods in efficient LLM pre-training remains underexplored. This work identifies three critical $\underline{O}$bstacles: ($O$1) lack of comprehensive evaluation, ($O$2) untested viability for scaling, and ($O$3) lack of empirical guidelines. To tackle $O$1, we summarize existing approaches into four atomic growth operators and systematically evaluate them in a standardized LLM pre-training setting. Our findings reveal that a depth-wise stacking operator, called $G_{\text{stack}}$, exhibits remarkable acceleration in training, leading to decreased loss and improved overall performance on eight standard NLP benchmarks compared to strong baselines. Motivated by these promising results, we conduct extensive experiments to delve deeper into $G_{\text{stack}}$ to address $O$2 and $O$3. For $O$2 (untested scalability), our study shows that $G_{\text{stack}}$ is scalable and consistently performs well, with experiments up to 7B LLMs after growth and pre-training LLMs with 750B tokens. For example, compared to a conventionally trained 7B model using 300B tokens, our $G_{\text{stack}}$ model converges to the same loss with 194B tokens, resulting in a 54.6% speedup. We further address $O$3 (lack of empirical guidelines) by formalizing guidelines to determine growth timing and growth factor for $G_{\text{stack}}$, making it practical in general LLM pre-training. We also provide in-depth discussions and comprehensive ablation studies of $G_{\text{stack}}$. Our code and pre-trained model are available at https://llm-stacking.github.io/.

## 1   Introduction

Emergent abilities of Large Language Models (LLMs) rely on scaling-up [1, 2]. Empirical evidence from scaling laws [3–5] fuels the development of increasingly larger models, pushing the boundaries of LLMs capabilities. However, pre-training these gigantic models comes at a significant cost in terms of energy consumption and environmental impact [6] (e.g., pre-training Llama-3 [7] consumes a total of 7.7M GPU hours and generates 2290 tons of carbon dioxide equivalent of carbon emissions). The efficient pre-training of LLMs is thus crucial, both from a scientific and a societal perspective, to ensure the continual growth and adoption of AI [8, 9].

One promising research direction to accelerate model training involves leveraging trained smaller (base) models to expedite the training of larger (target) models, a technique known as model growth.

---

[*] Equal Contributions.
[†] Work done during interning at HKUST.
[‡] Corresponding Author.

38th Conference on Neural Information Processing Systems (NeurIPS 2024).

Concretely, model growth studies how to leverage the trained smaller model's parameters $\Theta^{(s)}$ to initialize the larger model's parameters $\Theta^{(l)}$. Current popular methods generally focus on expanding the parameters of the base model through techniques like splitting [10–12], copying [13, 14], or matrix mapping [15]. There are also some approaches that initialize new parameters from scratch [16, 12, 17]. The primary objective is to accelerate the training of large models, and existing methods demonstrate promising speedup results on models such as BERT [11, 14, 18, 15, 12, 13]. Despite such empirical evidence and its alignment with the goal of efficient LLM pre-training, model growth methods are not widely adopted in the context of LLM pre-training [7, 19]. To our best knowledge, the only LLM that utilizes model growth for accelerating is FLM-101B [20], but it lacks a baseline LLM trained from scratch to compare. We observe three key Obstacles that hinder LLM pre-training from using existing model growth techniques, specifically:

• $O1$: Lack of comprehensive assessment. Some existing model growth methods report results on LLM pre-training, but either lack a baseline comparison [20] or are still in exploratory stages [15, 13]. In contrast, most growth approaches are evaluated in encoder-based BERT models [14, 11, 18, 12, 13, 16, 17], which have different architecture and training configurations compared to prominent decoder-based LLMs such as Llama [21].

• $O2$: The untested scalability. This scalability has two aspects: the model size and the amount of pre-training data. Regarding the model size, the existing approaches are only evaluated on smaller-scale BERT models or in preliminary experiments with LLMs. It is unclear whether these growth methods will continue accelerating training when applied to large-scale LLMs with more extensive evaluation. As for the amount of pre-training data, there are debates [22] over whether certain efficient training strategies may initially converge faster but ultimately perform similarly or worse than vanilla training methods when given ample computational resources (i.e., more training data).

• $O3$: Lack of empirical guidelines. Scaling laws [3, 4] give clear empirical guidelines on pre-training computational-optimized LLMs, greatly stimulating and advancing the field. Yet, there is a lack of empirical guidelines on growth techniques, discouraging LLM practitioners from adopting these approaches, especially considering the high costs of LLM pre-training.

These three obstacles are consequential in nature. Hence, in this work, we empirically revisit the concept of model growth as a solution to efficient LLM pre-training by tackling them one by one.

To tackle $O1$, we systematically evaluate model growth techniques on practical LLM pre-training. We first categorize existing growth methods and summarize them into four atomic growth operators, each of which can grow along two directions: widthwise (intra-layer) and depthwise (layer-wise). We illustrate them in Figure 2. These operators serve as representative choices for evaluating the performance of model growth techniques. We use these operators to expand 400M base models to 1.1B Llama-like LLMs and continually pre-train them. Next, we evaluate these growth techniques on the training loss and eight standard NLP benchmarks from the Harness toolkit [23]. We found the direct operator that stacks depthwisely $G_{stack}$ consistently outperforms others across overall evaluation metrics, demonstrating its potential in accelerating LLM pre-training. This motivates us to investigate extensively by addressing $O2$ and $O3$ on $G_{stack}$.

To address $O2$, we investigate the $G_{stack}$ operator's scalability to larger model sizes and to more training data. We conduct extensive experiments

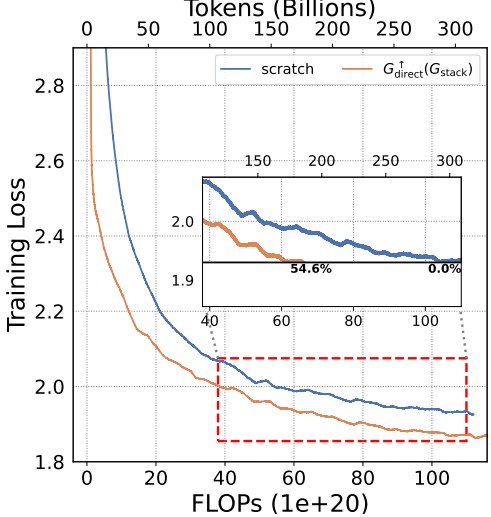

Figure 1: The training loss for two 7B LLMs, trained from scratch and with $G^{\uparrow}_{direct}$ ($G_{stack}$). At 300B tokens, $G_{stack}$ accelerates by 54.6% compared to scratch.

by scaling model size up to 7B parameters trained with 300B tokens, and pre-training a 410M model with over 750B training tokens. This is in contrast to the previous largest LLM pre-training experiment that uses model growth methods and has baselines for comparison, which is reported in Ligo [15], where a GPT2-1.5B model is trained for 15k steps (approximately 15B tokens). The

results are encouraging, as we consistently observe significant improvements $G_{stack}$ offers in both scenarios. For example, we achieve a remarkable 54.6% speedup in pre-training for a 7B model with 300B tokens (Figure 1). Interestingly, the loss improvement in our 750B-token experiment aligns with a logarithmic function. We further extend this logarithmic curve and determine that the improvement continues to be substantial even for the LLM trained with over 8T tokens. Moreover, we summarize all our experiments by estimating the LLM scaling law for LLMs pre-trained with $G_{stack}$. Given the same target loss value, our analysis reveals a significantly reduced computational cost compared to the common scaling law [4].

For $O3$, we explore the practical guidelines for using $G_{\text{stack}}$ in LLM pre-training. Given a computational budget, we determine the optimal strategy for two key factors of $G_{\text{stack}}$, growth timing $d$ and growth factor $g$. Growth timing $d$ relates to the training tokens used for small models before growing, and growth factor $g$ refers to the factor between the non-embedding parameter number of the large models and the small models. We formalize our findings into equations that offer concrete suggestions for utilizing $G_{\text{stack}}$. We believe this work could significantly pique the interest and bolster confidence in future LLM pre-training with model growth techniques, both in academia and industry.

To summarize, our contributions are four-fold: 1) We first systematically investigate model growth techniques and identify four atomic model growth operators, establishing a better understanding of the field in Section 3.1. 2) We then design a standard LLM pre-training testbed and perform comprehensive evaluations on these operators, finding that a simple depthwise stacking $G_{\text{stack}}$ exhibits significant superiority in Section 3. 3) We further demonstrate the scalability of $G_{\text{stack}}$ with experiments on LLMs ranging from 410M to 7B parameters and up to 750B training tokens in Section 4.1. 4) We also provide guidelines of equations on determining growth timing and growth factors for optimal use of $G_{\text{stack}}$ in Section 4.2.

## 2 Related Work - Model Growth for Efficient Pre-training

The idea of growing neural networks dates back to the 1990s [24–26]. The pioneering work of Net2Net [10] marks a milestone, for the first attempt to study model growth in deep learning era. Net2Net expands width and depth while keeping original functions (namely function preserving) via randomly splitting old neurons and injecting new identity layers. The widthwise splitting method of Net2Net represents a series of works that aim to "expand" the existing neurons to the desired larger size. Bert2Bert [11] serves as a BERT-based extension of the widthwise Net2Net. StagedGrow[13] doubles the width by concatenating two identical layers and halves final loss to keep function-preserving. Lemon [12] suggests integrating a parameter into the splitting of neurons in Bert2Bert, aiming to break weight symmetry. Depthwisely, StackedBert [14] simply stacks duplicated layers to form a deeper model. In contrast to the above direct copy/split approaches, LiGO [15]presents a learning-based method that initializes the larger model's parameters via learning a linear mapping from the smaller model's parameters.

Alongside the approaches that expand existing parameters, there are works that initialize new ones without relying on existing ones. For instance, MSG [17] proposes a multi-staged growing strategy that progressively expands transformer components, where the newly grown neurons are randomly initialized using a masking mechanism to ensure function preservation. Besides, some works have assigned specific values, like zero, to the newly initialized neurons to negate their influence [16, 12].

All the above methods are primarily explored in BERT or earlier stages of LLM pre-training. On the other hand, our objective is to present the first systematic review of model growth techniques in the LLMs era. To our knowledge, FLM-101B [20] is the only existing LLM that uses the growth method [17] for accelerating billion-scale LLM pre-training. Nonetheless, this work lacks a baseline model trained from scratch, making it difficult to assess the effectiveness of the model growth technique. In contrast, we aim to provide a comprehensive study by establishing a standardized testbed to compare LLMs trained from scratch and with various growth methods in LLM pre-training.

## 3 Systematically Assessing Model Growth for LLM Pre-Training

Existing model growth methods [14, 11, 18, 15, 12, 13, 16, 17] are mainly evaluated on BERT [27], with limited focus on decoder-only large-scale language models such as Llama [21]. Moreover, these growth methods are often not comparable due to different training settings [14, 11, 17, 12].

Even some growth LLMs experiments are evaluated, their results are often incomplete [20, 15]. To overcome these limitations, we first summarize existing works [14, 11, 18, 15, 12, 13, 16, 17] into four atomic growth operators to represent these growth techniques. Then we build a standardized LLMs training testbed to pre-train LLMs with four growth operators on depthwise and widthwise directions and evaluate the results with both training loss and eight evaluation metrics in Harness [23].

### 3.1 Growing LLMs with Growth Operators

Recent years, researchers have focused on enhancing the efficiency of training large models by making use of smaller pre-existing models [10, 11, 14, 18, 15, 12, 13, 16, 17]. These state-of-the-art methods can be categorized into two distinct groups. The first group focuses on deriving new neurons from the existing ones [10, 11, 14, 12, 15], while the second group focuses on initializing new parameters separately [18, 13, 16, 17]. Drawing from these two lines of research, we summarize four **atomic growth operators**. These operators include: **(A)** directly duplicating and stacking old layers in a depthwise manner or splitting neurons in the same layer widthwisely, denoted as $G_{\text{direct}}$, **(B)** generating expanded parameters using a learnable mapping matrix to the existing parameters, denoted as $G_{\text{learn}}$, **(C)** setting the new parameters to zero, denoted as $G_{\text{zero}}$, and **(D)** randomly initializing the new parameters, denoted as $G_{\text{random}}$. The illustration of four operators is shown in Figure 2. The $G_{\text{direct}}$ and $G_{\text{learn}}$ growth operators produce new neurons from the current ones, in contrast to $G_{\text{zero}}$ and $G_{\text{random}}$ which initialize new parameters independently. *For the formal definitions of the operators and the differences to the existing growth methods in design, please refer to Appendix A.* Complex growth methods, such as those involving auxiliary loss or exploring training dynamics like learning rates [28, 29, 16] are interesting. But considering the high computational cost of LLM pre-training, we focus on simple, universally applicable growth operators for different LLM pre-training settings.

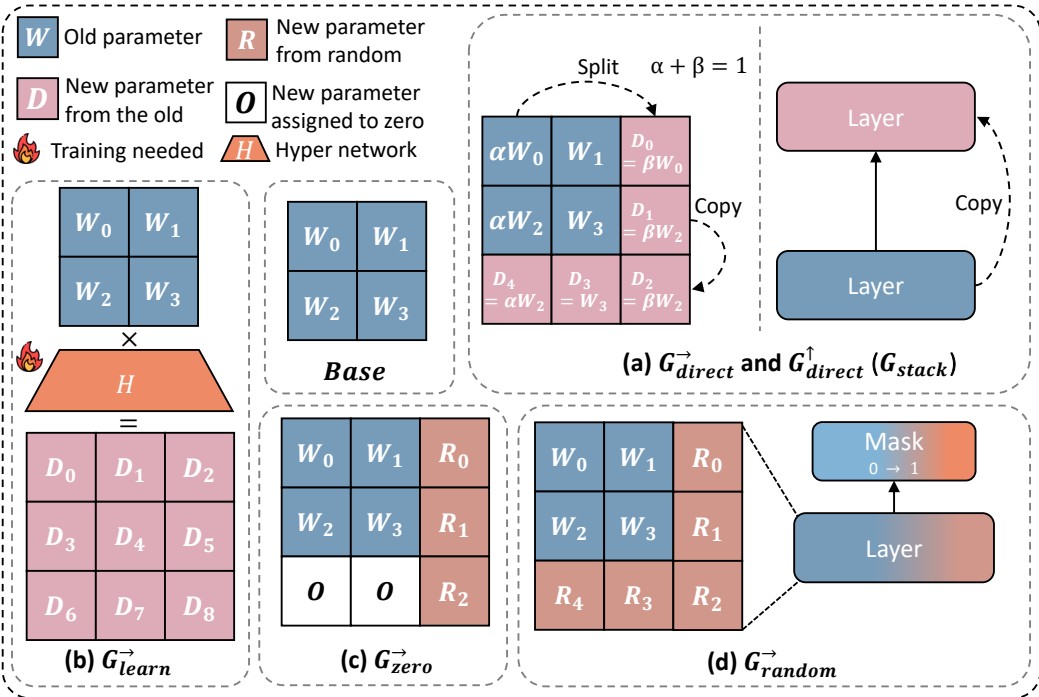

Figure 2: The simplified illustration of four growth operators $G_{\text{direct}}$, $G_{\text{learn}}$, $G_{\text{zero}}$ and $G_{\text{random}}$, each of which can grow along widthwise (intra-layer) $G^{\rightarrow}$ or depthwise (layer-wise) $G^{\uparrow}$. $\mathbf{W_n}$ is the parameters before growth, while $\mathbf{D_n}$, $\mathbf{R_n}$ and $\mathbf{O}$ are the growth parameters derived from the old, randomly initialized, and zero-initialized respectively. Except $G_{\text{direct}}$, other three operators only illustrates the widthwise growth.

To make a fair comparison of the four growth operators for LLM pre-training, we define a standardized "one-hop" growth process that involves two training phases, small model training before growth and large model training after growth. We first train the small LLMs with $d$ tokens before growing. Then, we use operator $G$ to grow them to the target LLMs by a factor of $g$ for non-embedding parameters

and then continual pre-training the large LLMs for $D$ tokens. Two key factors in the procedure are worth noting: the growth factor $g$ and the data for base model training $d$, which can be interpreted as "growth timing". We further evaluate each growth operator by separately examining in *depthwise (intra-layer)* growth $G^\uparrow$ and *widthwise (layer-wise)* growth $G^\rightarrow$. Concretely, we start with base models (400M LLMs) trained on $d = 10B$ tokens, apply the four operators in both directions to scale them up to the target size of 1.1B (approximately a growth factor of $g = 4$), and then continue training for an additional $D = 97.5B$ tokens. [4] Appendix B contains the LLM's architecture configuration and training details.

### 3.2 Pre-Training 1.1B LLMs

We report results on training loss, eight standard Harness NLP benchmarks along with the average accuracy and the speedup ratio in Figure 3. Our key discovery reveals that depthwise growth $G^\uparrow$ exhibits a significant acceleration over both widthwise growth $G^\rightarrow$ and training models from scratch, while surprisingly, $G^\rightarrow$ does not offer any notable advantages. Among the depthwise growth operators, $G^\uparrow_{direct}$, $G^\uparrow_{learn}$, and $G^\uparrow_{zero}$, all outperform the baseline and $G^\uparrow_{random}$. The underperformance of $G^\uparrow_{random}$ in our study may be attributed to its design for gradual "mini-step" growth [17], whereas our unified approach uses a single step. Most notably, **depthwise stacking $G^\uparrow_{direct}$ emerges as the clear winner among growth operators, surpassing its competitors in speedup, training loss and nearly every Harness evaluation metric**. For example, compared to training models from scratch for 100B tokens, $G^\uparrow_{direct}$ achieves a significant efficiency gain, increasing training speed by 49.1%. The calculation of speedup please refer to Appendix B.2. The Appendix C presents more experiments on these operators, including their loss training and evaluation figures.

| | Depth | | | | Width | | | | Baseline |
| --- | --- | --- | --- | --- | --- | --- | --- | --- | --- |
| | $G^\uparrow_{direct}$ | $G^\uparrow_{zero}$ | $G^\uparrow_{random}$ | $G^\uparrow_{learn}$ | $G^\rightarrow_{direct}$ | $G^\rightarrow_{zero}$ | $G^\rightarrow_{random}$ | $G^\rightarrow_{learn}$ | $scratch$ |
| Lambada ($\uparrow$) | 48.20 | 48.67 | 44.14 | 48.36 | 46.16 | 44.67 | 44.24 | 45.66 | 47.87 |
| ARC-c ($\uparrow$) | 29.18 | 28.32 | 28.41 | 27.38 | 28.58 | 26.70 | 27.64 | 26.70 | 27.21 |
| ARC-e ($\uparrow$) | 54.25 | 51.76 | 52.69 | 51.17 | 51.55 | 49.70 | 53.82 | 50.37 | 48.86 |
| Logiqa ($\uparrow$) | 28.87 | 27.95 | 25.96 | 28.11 | 27.34 | 25.03 | 26.11 | 26.57 | 25.96 |
| PIQA ($\uparrow$) | 71.98 | 71.81 | 70.78 | 71.16 | 69.47 | 69.74 | 70.13 | 69.91 | 69.64 |
| Sciq ($\uparrow$) | 81.1 | 81.9 | 77.7 | 80.0 | 81.4 | 76.0 | 79.5 | 79.5 | 76.8 |
| Winogrande ($\uparrow$) | 56.03 | 56.98 | 53.35 | 54.45 | 54.22 | 54.93 | 52.95 | 53.51 | 54.53 |
| Avg. ($\uparrow$) | 52.80 | 52.48 | 50.43 | 51.52 | 51.25 | 49.54 | 50.63 | 50.32 | 50.12 |
| Wikitext ($\downarrow$) | 16.73 | 17.35 | 17.85 | 16.93 | 18.03 | 18.76 | 18.29 | 18.44 | 17.98 |
| Loss ($\downarrow$) | 2.151 | 2.161 | 2.258 | 2.156 | 2.209 | 2.249 | 2.227 | 2.233 | 2.204 |
| Speed-up ($\uparrow$) | 49.1% | 46.6% | -25.7% | 48.6% | -0.7% | -17.9% | -13.8% | -15.4% | 0.0% |

Figure 3: We evaluate operators using training loss and Lambada [30], ARC-c [31], ARC-e [31], Logiqa [32], PIQA [33], Sciq [34], Winogrande [35] and Wikitext PPL [36] totaling eight standard NLP benchmarks. After $8 \times 10^{20}$ FLOPs of training, $G^\uparrow_{direct}$ demonstrates a significant speedup.

## 4   Delving Deeper Into Depthwise Stacking ($G_{stack}$)

The empirical evidence suggests that certain growth operators, most notably $G^\uparrow_{direct}$, exhibit an impressive acceleration in LLM pre-training compared to the baseline approach of training models from scratch. We now turn our attention to a more in-depth examination of the $G^\uparrow_{direct}$. For ease of reference, **we will henceforth denote this depthwise stacking approach as operator $G_{stack}$**:

---

[4]Given growth factor $g = 4$, the sum of FLOPs for training $d = 10B$ and $D = 97.5B$ approximately equals to consumption for training large LLMs $D = 100B$, which is the FLOPs of our baseline trained from scratch.

$\mathcal{M} = \underbrace{M \circ M \circ \cdots \circ M}_{g \times M}$, where $M$ is a small base model trained with $d$ tokens, $\mathcal{M}$ is the target model and $g$ is the growth factor.

This section addresses the two main challenges (*O2* and *O3*) outlined in the introduction: 1) evaluating the performance of $G_{\text{stack}}$ in scaling scenarios, i.e. larger model sizes and more training tokens; and 2) determining the hyperparameters when using $G_{\text{stack}}$, i.e., the growth timing $d$ and growth factor $g$.

## 4.1 Scaling $G_{\text{stack}}$

**Scaling Model Sizes for $G_{\text{stack}}$.** Our scaled-up experiments involve two larger model sizes: 3B and 7B. We initially train smaller models with a layer count that is one-quarter of our target layers (growth factor $g = 4$), utilizing 10B tokens ($d = 10B$). Then, we train the stacked models using over 300B tokens ($D = 300B$) for both sizes. Figures 4 and 5 show the loss, and the NLP benchmarks average accuracy evaluated using the Harness evaluator for training 3B and 7B LLMs with 300B tokens, respectively.[5] The acceleration of $G_{\text{stack}}$ is consistent across two

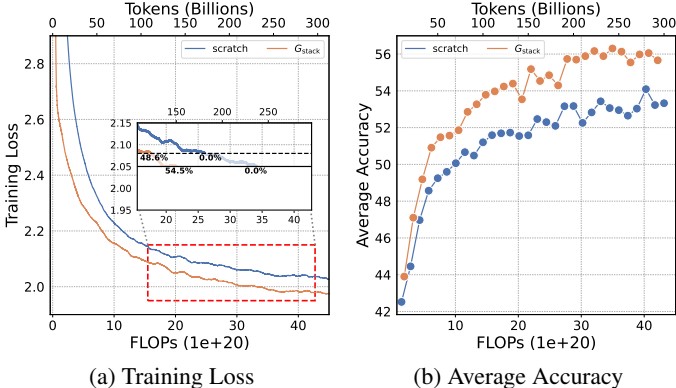

(a) Training Loss        (b) Average Accuracy

Figure 4: Training 3B LLMs with 300B tokens. $G_{\text{stack}}$ significantly outperforms scratch in (a) loss and (b) average accuracy across NLP benchmarks. At 180B and 240B tokens, $G_{\text{stack}}$ accelerates by 48.6% and 54.5% compared to scratch.

models and all evaluation metrics. For instance, considering the 3B model, Figure 4 demonstrates that $G_{\text{stack}}$ achieves a 54.5% speedup in pre-training, improvement of 2.1 in NLP benchmarks average accuracy compared to the baseline 3B model trained with 240B tokens.

When comparing the 1B, 3B, and 7B models, it is evident that the benefits of $G_{\text{stack}}$ are not reduced as the model size increases, implying that its acceleration effect can be leveraged even with larger models. Details of the evaluation results, including evaluation with instruction tuning, can be found in Appendix D. Appendix E compares our baselines with the open-source LLMs Pythia and tinyLlama.

**Scaling Training Tokens for $G_{\text{stack}}$.** We next evaluate the scalability of the stacking operator on another dimension - training with more tokens. This is especially important in light of recent discussions about the validity of efficient training algorithms, which have sparked debate [22] over whether certain strategies may initially learn faster but ultimately perform similarly or worse than vanilla training methods when given more training data. Hence, we aim to pre-train a LLM using $G_{\text{stack}}$ on a substantial amount of training tokens.

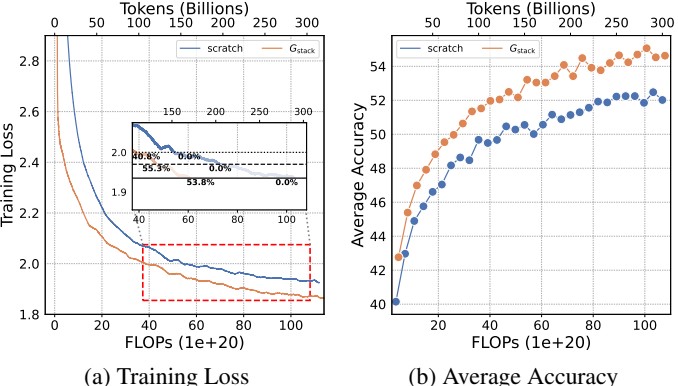

(a) Training Loss        (b) Average Accuracy

Figure 5: Training 7B LLMs with 300B tokens. $G_{\text{stack}}$ significantly outperforms scratch in (a) loss and (b) average accuracy across NLP benchmarks. At 160B, 220B and 280B tokens, $G_{\text{stack}}$ accelerates by 40.8%, 55.3% and 53.8% compared to scratch.

---

[5]In this study, we always calculate the consumption by combining the FLOPs required for both training small models and large models. So given $g = 4$, the consumption for training small model $d = 10B$ equals to the cost for training $D = 2.5B$, so the plotted curves for $G_{\text{stack}}$ actually starts at $2.5B$.

Concretely, we conduct an experiment on a 410M LLM using 750B tokens. Following the experimental setup in the previous section, we set growth ratio $g = 4$ and growth timing $d = 10B$ and conduct continuous pre-training on the target 410M LLMs for 750B tokens. Compared to the chinchilla-recommended 8B tokens [4] for the 410M model, our experimental setting also surpasses this value by nearly 100 times, reaching 750B tokens.

The training dynamics on Figure 6a indicate that $G_{\text{stack}}$ remains effective in such cases. Details of the evaluation results with the similar findings can be

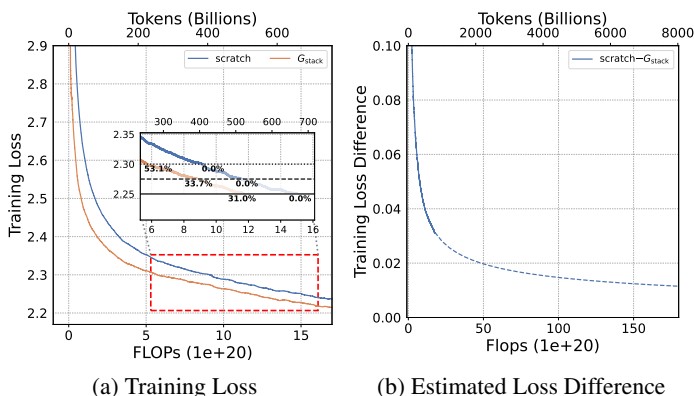

(a) Training Loss    (b) Estimated Loss Difference

Figure 6: Training 410M LLMs with 750B tokens. $G_{\text{stack}}$ significantly outperforms scratch in (a) loss. At 400B tokens, we observe a 53.1% acceleration, and even at 700B tokens, there is still a 31.0% acceleration. (b) We fit the difference between the losses of the scratch and $G_{\text{stack}}$ and find that the acceleration with $G_{\text{stack}}$ remain sustainable for longer training.

found in Appendix D.3. Building upon the exceptional stability of LLM pre-training [37, 38], we estimate loss improvements and plot them in Figure 6b. The fitting curve indicates $G_{\text{stack}}$ will continue to exhibit acceleration effects even after 8T tokens, which is over 1000 times longer than the recommended token number [4]. It is also notable that this loss improvement after 8T training is not trivial for LLM pre-training, as previous studies [39] suggest that even minor improvements in the later phase can have a relatively substantial impact on downstream performance.

From a LLM practitioner's perspective, this is also crucial considering "overtraining", which involves training LLMs with significantly larger amounts of data than recommended by scaling laws [3–5], a common practice that has become prevalent. A notable example is the training of LLama 3-8B with 15T tokens, which is nearly 100 times greater than the token count recommended by the chinchilla scaling laws [4]. Hence, this finding provides confidence in the consistent excellent acceleration of $G_{\text{stack}}$ throughout the entire practical LLM pre-training process.

**Estimating Scaling Laws.** To further explore our findings, we graph our four models (410M, 1.1B, 3B, and 7B) on the same figure and attempt to uncover our "scaling law" using the $G_{\text{stack}}$ operator. Following [3, 4], we define the scaling power law using the equation $L_C = aC^b$, where $a$ and $b$ are constants we need to fit, $C$ represents the FLOPs, and $L_C$ denotes the model's final loss under this FLOP. We use the curve_filt function in SciPy [40] to fit both the scratch model and the $G_{\text{stack}}$ model and present the estimation scaling law in Figure 7. The figure shows that our $G_{\text{stack}}$ scaling law exhibits improved efficiency compared to the scaling law estimated from baseline LLMs, achieving the same target loss while requiring much less computational resources. However, in light of the significant computational resources devoted to other scaling law studies [3, 4], we acknowledge that our $G_{\text{stack}}$ scaling law is an initial estimate subject to computation constraints, and a comprehensive study is left for future research.

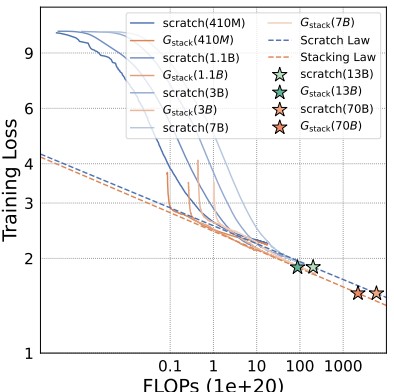

Figure 7: We plot scaling law lines based on 410M, 1.1B, 3B, 7B LLMs and make two predictions at the same losses of original computational-optimized 13B and 70B LLMs.

## 4.2 Determining Growth Timing and Growth Factor for Using $G_{\text{stack}}$

We comprehensively validate the effectiveness of the $G_{\text{stack}}$ compared to training from scratch in Section 4.1. However, to incorporate $G_{\text{stack}}$ into a LLM's pre-training process, we need to determine two crucial hyperparameters: the growing time ($d$) and the growing factor ($g$). In our previous experiments, we rely on ad-hoc choices for these parameters, thereby lacking a systematic approach

to determining them when use $G_{\text{stack}}$. There exists research on investigating the growth timing [41], but the settings are quite different from the LLM pre-training. Therefore, this section offers a clear guide for practitioners looking to optimize using the $G_{\text{stack}}$ operator in LLM pre-training processes.

We begin by offering a formal definition. When given a computational budget $C$, established scaling power laws [3, 4] exist to guide the non-embedding parameters $N$ and the number of training tokens $D$ to achieve the lowest model loss in the case of training from scratch. However, tuning hyperparameters becomes more complex when the fixed budget $C$ is allocated to find the optimal model training strategy using the $G_{\text{stack}}$ operator, which involves two training phases. Consequently, the overall computational budget $C$ can be expressed as the sum of the two components: $C = C1 + C2$. Here, $C1$ and $C2$ represent the flops required to train the initial small models $C1 = FLOPs(n, d)$, and the large model $C2 = FLOPs(N, D)$ respectively, where $n$ and $d$ denote the parameters and training tokens of the small model, and $N$ and $D$ represent the parameters and training tokens of the large model. Since the large model is grown by a factor of $g$ such that $N = gn$, we have $C = C1 + C2 = FLOPs(g, N, d) + FLOPs(N, D) = FLOPs(g, N, d, D)$.

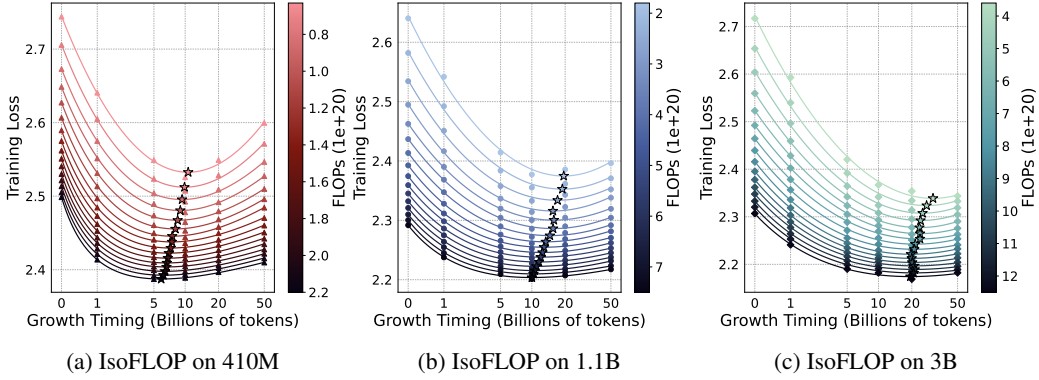

|     (a) IsoFLOP on 410M     |     (b) IsoFLOP on 1.1B     |     (c) IsoFLOP on 3B     |

Figure 8: In 410M, 1.1B, and 3B LLMs, we plot smoothed loss curves for different growth timing $d$ given a set of FLOPs to form IsoFLOP figures. We find a clear valley in loss, indicating that for a given FLOP budget, there exists an optimal growth timing $d$ for the $G_{\text{stack}}$ operation.

So when given a budget $C$, our objective is to identify the optimized values $D$, $N$, $d$, $g$ that minimize the loss $\mathcal{L}(D, N, d, g)$. However, simultaneously optimizing the above four variables can be computationally expensive. Therefore, instead of searching for global optimals, we separately determine two factors closely related to the $G_{\text{stack}}$: the training tokens for the small model (growth timing) $d$ and the growth factor $g$:

$$\arg\min_{f,h} \mathcal{L}(D, N, d, g), \quad \text{where } d = f(D, N), g = h(D, N) \tag{1}$$

**Determining Growth Timing:** $d$. We first explore the effect of growth timing, i.e. the training token $d$ for the small model. Particularly, we apply the $G_{\text{stack}}$ operator to a series of small models trained with $d = 0B, 1B, 5B, 10B, 20B, 50B$ tokens. Subsequently, we stack them to the target layers with growth factor $g = 4$ and train for a fixed set of computational FLOPs. We replicate the above experiments using three target model sizes $N = 410M, 1.1B, 3B$ and plot each set of IsoFLOP points in Figure 8a, 8b and 8c. Surprisingly, even a small model trained with just $1B$ tokens exhibits a significant speedup compared to the directly stacking small random initialized models (represented as "0B"). While 0B's performance is similar to models trained from scratch, implying stacking itself does not serve as an effective initialization method. Furthermore, by applying smoothing techniques to model IsoFLOP curves as parabolas, we identify the optimized value of $d$ that minimizes loss for each FLOP count, leading us to hypothesize the existence of a logarithmic equation involving $N$, $C$, and $d$:

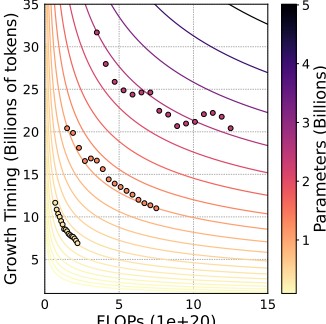

Figure 9: We fit a contour figure for predicting $d$ given $C$ and $N$. These optimal growth timing $d$ fit the figure well.

$$log_{10}(d) = a \log_{10}(N) + \frac{b}{\log_{10}(C)} + c \qquad (2)$$

After fitting, we obtain $a = 0.88$, $b = 163.27$ and $c = -5.74$ and we plot the contour figure in Figure 9. It can be observed that our estimated curves align well with the actual optimal points.

**Determining Growth Factor:**
$g$. Another factor we determine is the growth factor $g$. As models with 3B and 7B parameters have identical depths, we run experiments using two model sizes: 1.1B (24 layers) and 3B (32 layers). Specifically, we vary the stack factors to $g = 2, 4, 8, 24$ for the 1.1B model and $g = 4, 8, 16, 32$ for the 3B model while keeping the base models trained with $d = 10$B tokens. The smoothed IsoFLOP curves are plotted in Figure 10. Interestingly, even with a relatively shallow 2-layer base model and a growth factor

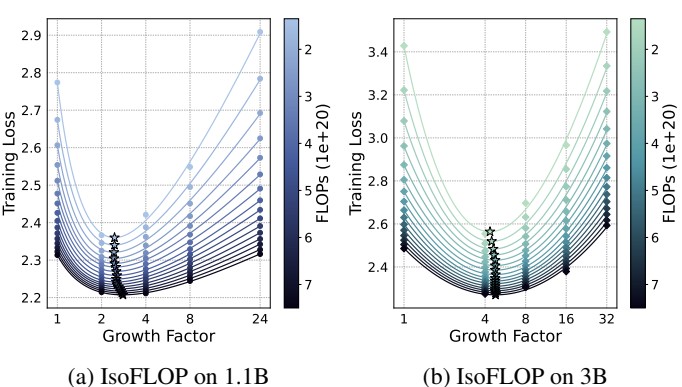

(a) IsoFLOP on 1.1B    (b) IsoFLOP on 3B

Figure 10: In 1.1B, and 3B LLMs, we plot smoothed loss curves for different growth factor $g$ given a set of FLOPs as IsoFLOP figures. The optimal g falls between 2 and 4.

of $g = 16$, we observe a remarkable improvement compared to the baseline 3B model ($g = 1$). However, when using a 1-layer base model, $G_{\text{stack}}$ underperforms compared to the baseline. Our curves indicate that the optimal growth factor $g$ lies between 2 and 4.

However, unlike determining training token $d$, we cannot generate sufficient data to estimate the relationship between $N$, $C$, and $g$, due to computational constraints. Thus, this work suggests a constant growth factor of $g = 4$. We also include our preliminary estimated equation and contour figure for $g$ in the Appendix F. All evaluation results of Section 4.2 are listed in Appendix G.

## 5 Ablation and Discussion

To further give insights into adopting model growth techniques in LLM pre-training, we ablate variances for $G_{\text{stack}}$ and discuss function preserving in general model growth techniques.

### 5.1 Ablation: How to Stack?

It is worth noting that $G_{\text{stack}}$ differs from the algorithm proposed in StackedBERT [14], which utilizes a gradually stacking strategy. Hence, we compare our "one-hop" $G_{\text{stack}}$ and their gradual stacking approach. Following the methodology introduced in StackBERT, we employ a two-step stack strategy. Given our target model size of 1.1B with 24 layers, we start with a 6-layer model. Subsequently, we train it on 10B tokens and double the model's depth through stacking, repeating this step twice (train-stack-train-stack) to achieve the desired scale. Our experiments demonstrate that $G_{\text{stack}}$ outperforms gradual stacking approaches on loss and downstream evaluations. For example, the evaluation results show that $G_{\text{stack}}$ achieves a 2.4 higher average accuracy and 0.6 better Wikitext PPL than gradual stacking when pre-training large models for 100B tokens. The results can be found in Appendix H.1. We further compare other stacking variations, such as stacking via interpolation and partial stacking of certain layers which are also adopted in LlamaPro [42] and Solar [43], and leave our detailed findings in the Appendix H.2 and H.3.

### 5.2 Discussion: Why Does Function Preserving Fail?

Function preservation (FP) is a key concept that underlies most model growth approaches [10–12, 17]. The idea is intuitive that a larger model should initialize parameters that can represent the same

function as the ones in the smaller model, i.e. $\forall x, f(x; \Theta^{(s)}) = f(x; \Theta^{(l)}_{init})$, where $x$ is the input. We give a mathematical definition of FP in the Appendix I.1.

We find it intriguing that our $G_{\text{stack}}$ approach, which violates FP, emerges as the most effective in our study. To further investigate, we conduct a simple ablation study to break FP by introducing noise on the strict-FP operator $G_{\text{direct}}^{\rightarrow}$. We initialize the new neurons by a weighted combination of two sets of parameters: those from $G_{\text{direct}}^{\rightarrow}$ and those from random initialization. The weighting factor is controlled by a noise ratio. Our findings are intriguing. After 40B tokens training, adding 20% noise outperforms original $G_{\text{direct}}^{\rightarrow}$ by 0.27 on the Wikitext PPL and 0.41 on the average accuracy score.

We also add noise for $G_{\text{stack}}$. When we add 20% noise, our LLM performs slightly better than the no-noise model. However, when the noise level exceeds 20%, the performance significantly deteriorates. These results indicate that function preservation may not be the sole determining factor for model growth. **In other words, exploring ways to accelerate the training of larger models and strict preserving function during growth might represent two overlapping yet distinct research directions.** The experimental details are provided in the Appendix I.2.

# 6 Conclusion

This work empirically explores model growth approaches for efficient LLM pre-training. We address three key challenges of current model growth research for efficient LLM pre-training. We first comprehensively evaluate model growth techniques into four atomic operators and explore depthwise growth $G_{\text{stack}}$ beats all other methods and baselines in various evaluations. We next address concerns about the scalability of $G_{\text{stack}}$ by extending the model and training data scales. Furthermore, we systematically analyze the usage of the $G_{\text{stack}}$ operator, focusing on growth timing and growth factor. Based on this analysis, we formalize a set of guidelines for effectively utilizing the $G_{\text{stack}}$ operator. In addition, we provide in-depth discussions and comprehensive ablation studies of $G_{\text{stack}}$, shedding light on the broader implications of our work.

# 7 Limitations

While our work has demonstrated remarkable potential, four limitations deserve further attention. One limitation is the constraint of computation resources. For example, we only compare two sets of growth factor $d$ configurations, which limits the capacity to derive a formula for determining the optimal growth factor $d$. Another limitation of our work is the focus on relatively simple operator choices, where we prioritize simplicity over exploring more sophisticated strategies. For instance, we do not extensively investigate the multi-step growth or dynamic modifications to the training process, such as adjusting the learning rate during continual pre-training. The third limitation involves the incomplete cosine learning rate schedule during training. This also arises from the resource-intensive nature of pre-training LLMs and the constraints on available computational resources. Therefore, we adopt a strategy where we initially set a large number of training tokens and then we pre-train LLMs until the training runs are interrupted by tasks with higher priority. Lastly, although this study's scope is an empirical exploration and the content is self-contained, there is a lack of theoretical insights into the success of $G_{\text{stack}}$ in LLM pre-training.[6] Nonetheless, we will release all LLM checkpoints to facilitate the community's investigation into the theoretical principles behind our observations.

# 8 Acknowledgments

We thank all constructive comments from anonymous reviewers. Reynold Cheng and Wenyu Du were supported by the Hong Kong Jockey Club Charities Trust (Project 260920140), the University of Hong Kong (Project 109000579), the HKU Outstanding Research Student Supervisor Award 2022-23, and the HKU Faculty Exchange Award 2024 (Faculty of Engineering).

---

[6]A very recent paper indicates training LLMs via stacking may improve in reasoning [44].

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

# A    Details of Growth Operators

## A.1    Four Growth Operators

### A.1.1    Operator $G_{\text{direct}}$: Direct Derivation of Grown Parameters From Old Parameters

One intuitive strategy for expanding neural networks involves directly duplicating or splitting existing neurons. [14, 11, 12]. Unlike other growth operators, we distinguish between growth in terms of depth and width.

For width-wise expansion, the Net2Net technique and its transformer implementations [10, 11] involve splitting old neurons into two or more parts, with each splitting step achieving a=b+c. Depending on the specific splitting mechanism, there are two variations: even splitting and uneven splitting. The latter is proposed to address symmetry issues that arise when neurons are evenly split. In this paper, we adopt the approach of uneven splitting.

In the context of depth-wise expansion, a common practice is to duplicate layers, often referred to as "stacking" [14]. Therefore, we use the term $G_{\text{stack}}$ to represent this operator. While this approach may appear to deviate from function preservation, it surprisingly yields a strong baseline.

### A.1.2    Operator $G_{\text{learn}}$: Generation of New Parameters through Matrix Transformation

$G_{\text{learn}}$ is an operator that learns a matrix transformation function to map small models to a larger one [15]. This operator is applicable to both width and depth expansion. Considering the original model $f$ with parameters $\theta$, the target model $F$ with parameters $\Theta$, and $G_{\text{learn}}$ as the hypernetwork for meta-learning, the training corpus is denoted as $\mathcal{D}$, and the language model loss is denoted as $\mathcal{L}$. Then, we optimize the following objective:

$$\underset{G_{\text{learn}}}{arg\ min}\ \mathbb{E}_{x \sim \mathcal{D}}\ \mathcal{L}(x; F_\Theta), \quad \text{where } \Theta = G_{\text{learn}}(\theta) \tag{3}$$

### A.1.3    Operator $G_{\text{zero}}$: Setting New Parameters to 0

Setting new parameters to zero is often considered a simple method to achieve function preservation. However, optimizing networks with a significant number of zeros can present challenges. To tackle this issue, we adopt current practices that selectively zero out either the fan-in or fan-out parameters [13, 16, 12]. Specifically, for operator $G_{\text{zero}}$, during width growing, we zero out only the set of fan-out parameters for new neurons and randomly initialize the remaining ones. In the case of depthwise expansion, we zero out the final output layer of the newly-duplicated transformer blocks' MultiHead Attention and MLP.

### A.1.4    Operator $G_{\text{random}}$: Random Initialization of New Parameters

This group follows the common practice of randomly initializing new parameters. In earlier attempts, old neurons were frozen after the growth process [18, 17]. However, to ensure function preservation, a recent study introduces a mask for new neurons after expansion [17]. This mask is gradually removed during ongoing training. We refer to this new approach as the growth operator $G_{\text{random}}$.

## A.2    Difference of Our Operators and Base Methods

The operators $G_{\text{direct}}^{\rightarrow}$ shares a similar setting to Lemon with minor variances due to Llama achitectures. $G_{\text{learn}}$ is consistent with the methods LiGO, but with our own implementation. For $G_{\text{zero}}$, our approach aligns with Lemon in terms of depth, but differs from stagedTraining in width. Unlike stagedTraining, we do not double the width and assign zeros to the off-diagonal entries. Instead, our approach is more flexible; by zeroing out the submatrix in the bottom-left corner, we can extend it to any dimension. Our $G_{\text{random}}$ does not exhibit the "multi-hop" growth like MSG, instead, it grows "one-hop" directly to the target size. Our implementation of $G_{\text{direct}}^{\uparrow}$ ($G_{\text{stack}}$) differs from the algorithm employed in stackedBert. In stackedBert, a gradual growing technique is utilized, whereas our operator follows a more direct approach.

## A.3 Details of $G_{\text{direct}}$

**Embedding** Consider $E \in \mathbb{R}^{V \times d}$, and our goal is to expand it to $E' \in \mathbb{R}^{V \times D}$, $G_{\text{direct}}$ just copy some columns:

$$E' = G_{\text{direct}}(E) \tag{4}$$
$$= ER \tag{5}$$
$$= E \left[ \underbrace{\begin{matrix} I & & I \\ & I & \end{matrix}}_{d} \right] \tag{6}$$

where $R \in \mathbb{R}^{d \times D}$ is used to copy the embedding matrix $E$.

**Linear** Consider $W \in \mathbb{R}^{d_{out} \times d_{in}}$, target parameter $W' \in \mathbb{R}^{D_{out} \times D_{in}}$, where $d_{out} \leq D_{out}, d_{in} \leq D_{in}$, $G_{\text{direct}}$ is defined as:

$$W' = G_{\text{direct}}(W) \tag{7}$$
$$= LWR \tag{8}$$
$$= d_{out} \left\{ \begin{bmatrix} I \\ I \end{bmatrix} I \right] W \left[ \alpha \underbrace{\begin{matrix} I \end{matrix}}_{d_{in}} \beta \right] \tag{9}$$

where $R \in \mathbb{R}^{d_{in} \times D_{in}}$ is used for expanding the fan-in and $L \in \mathbb{R}^{D_{out} \times d_{out}}$ is used for expanding the fan-out. To satisfy function preserving, we ensure that $\alpha + \beta = I$.

**RMSNorm** For RMSNorm, a similar approach is adopted, consider parameter $\mu \in \mathbb{R}^d$, expanded parameter $\mu' = \frac{\sqrt{d}}{\sqrt{D}}[\mu, \ \mu_{0,D-d}] \in \mathbb{R}^D$:

$$RMSNorm'(x') = \frac{x'}{\sqrt{\frac{1}{D} \sum_{i=1}^{D} x_i'^2}} \odot \mu' \tag{10}$$

$$= [\sqrt{\frac{\sum_{i=1}^{d} x_i^2}{\sum_{i=1}^{D} x_i'^2}} \times RMSNorm(x), \ \zeta] \tag{11}$$

Therefore, using the $G_{\text{direct}}$, it is not possible to achieve function preservation for RMSNorm

**Depth ($G_{\text{stack}}$)** Consider a transformer with $l$ layers represented as $F = f_0 \circ f_1 \circ \cdots \circ f_l$. Our objective is to expand it to $L$ layers, where $L$ is a multiple of $l$. We have various stacking forms for this purpose, such as (a) direct stacking: $F' = F \circ F \circ \cdots \circ F$.

---

**Algorithm 1** Operator $G_{\text{stack}}$

---

**Input:** Base model $M_k^l$ with $l$ layers trained using dataset $d_k$ where $k$ is iteration steps. Growth factor $g$.
**Output:** Target Model $\mathcal{M}_0^{gl}$ with $gl$ layers
$\mathcal{M}_0^l = M_k^l$
**for** $t = 2 \ to \ g$ **do** $\quad\quad\quad\quad\quad\quad\quad\quad\quad\quad\quad\quad\quad\quad\quad\quad\quad\quad$ ▷ Model Stacking
$\quad \mathcal{M}_0^{tl} = \mathcal{M}_0^{(t-1)l} \circ M_k^l$
**end**

---

## A.4 Details of $G_{\text{zero}}$

**Embedding**   Consider an embedding matrix $E \in \mathbb{R}^{V \times d}$. The $G_{\text{zero}}$ operator expands it to $E' \in \mathbb{R}^{V \times D}$ with $O$, where $d \leq D$. Formally:

$$E' = [E, \ O] \tag{12}$$

Therefore, give a token $x$, the expanded embedding can be expressed as:

$$Embedding'(x) = \mathbb{1}_x E' = [Embedding(x), \ 0_{D-d}] \tag{13}$$

**Linear**   Consider parameter $W \in \mathbb{R}^{d_{out} \times d_{in}}$. $G_{\text{zero}}$ expand it to $W' \in \mathbb{R}^{D_{out} \times D_{in}}$, where $d_{out} \leq D_{out}$ and $d_{in} \leq D_{in}$. Formally:

$$W' = \begin{bmatrix} W & \mathcal{A} \\ O & \mathcal{C} \end{bmatrix} \tag{14}$$

where $\mathcal{A}, \mathcal{C}$ are randomly initialized new parameters. Considering the input token $x \in \mathbb{R}^{d_{in}}$ before expansion, and the input after expansion $x' \in \mathbb{R}^{D_{in}}$:

$$x' = [x, \ 0_{D_{in}-d_{in}}] \tag{15}$$

$$
\begin{aligned}
Linear'(x') \ &= \ x'W'^T \tag{16} \\
&= \ [x, \ 0_{D_{in}-d_{in}}] \begin{bmatrix} W^T & O \\ \mathcal{A}^T & \mathcal{C}^T \end{bmatrix} \tag{17} \\
&= \ [xW^T, \ 0_{D_{out}-d_{out}}] \tag{18} \\
&= \ [Linear(x), \ 0_{D_{out}-d_{out}}] \tag{19}
\end{aligned}
$$

**RMSNorm**   Considering the parameter $\mu \in \mathbb{R}^d$, $G_{\text{zero}}$ expand it to $\mu' = [\alpha\mu, \ \xi]$ like $G_{\text{random}}$ in Appendix A.5, because the input must be $x' = [x, \ 0_{D-d}] \in \mathbb{R}^D$.

**Depth**   In depth, by retaining only the residual part and initializing the MHA and SwiGLU final linear projections to zero, the MHA and SwiGLU layers can achieve function preservation.

## A.5 Details of $G_{\text{random}}$

**Embedding**   Consider an embedding matrix $E \in \mathbb{R}^{V \times d}$. The goal of $G_{\text{random}}$ is to expand it to $E' \in \mathbb{R}^{V \times D}$, where $d \leq D$. Formally:

$$E' = [E, \ \mathcal{E}] \tag{20}$$

where $\mathcal{E} \in \mathbb{R}^{V \times (D-d)}$ represents randomly initialized new parameters. We use a mask $c \in \mathbb{R}^D$ to mask out the randomly initialized parts:

$$c = [1_d, \ 0_{D-d}] \rightarrow [1_d, \ 1_{D-d}] \tag{21}$$

Therefore, for a token $x$, the masked embedding can be expressed as:

$$Embedding'(x) = \mathbb{1}_x E' \odot c = [Embedding(x), \ 0_{D-d}] \tag{22}$$

**Linear**  Consider parameter $W \in \mathbb{R}^{d_{out} \times d_{in}}$. Our goal is to expand it to $W' \in \mathbb{R}^{D_{out} \times D_{in}}$, where $d_{out} \leq D_{out}$ and $d_{in} \leq D_{in}$. Formally:

$$W' = \begin{bmatrix} W & \mathcal{A} \\ \mathcal{B} & \mathcal{C} \end{bmatrix} \tag{23}$$

where $\mathcal{A}, \mathcal{B}, \mathcal{C}$ are randomly initialized new parameters. Considering the input token $x \in \mathbb{R}^{d_{in}}$ before expansion, and the input after expansion $x' \in \mathbb{R}^{D_{in}}$:

$$x' = [x,\ 0_{D_{in}-d_{in}}] \tag{24}$$

$$x'W'^{T} = [x,\ 0_{D_{in}-d_{in}}] \begin{bmatrix} W^{T} & \mathcal{B}^{T} \\ \mathcal{A}^{T} & \mathcal{C}^{T} \end{bmatrix} \tag{25}$$

$$= [xW^{T},\ x\mathcal{B}^{T}] \tag{26}$$

To ensure that the expanded part of $x'$ starts with zeros, we still utilize a mask:

$$c = [1_{d_{out}},\ 0_{D_{out}-d_{out}}] \rightarrow [1_{d_{out}},\ 1_{D_{out}-d_{out}}] \tag{27}$$

$$Linear'(x') = x'W'^{T} \odot c = [Linear(x),\ 0_{D_{out}-d_{out}}] \tag{28}$$

**RMSNorm**  Considering the parameter $\mu \in \mathbb{R}^{d}$, our objective is to expand it to $\mu' = [\alpha\mu,\ \xi] \in \mathbb{R}^{D}$, where $\alpha$ is an undetermined coefficient and $\xi$ is a randomly initialized new parameter. Let the input be $x' = [x,\ 0_{D-d}] \in \mathbb{R}^{D}$, then we have:

$$\sum_{i=0}^{D} x'^{2} = \sum_{i=0}^{d} x^{2} \tag{29}$$

$$RMSNorm'(x') = \frac{x'}{\sqrt{\frac{1}{D}\sum_{i=0}^{D} x_i'^{2}}} \odot \mu' \tag{30}$$

$$= \frac{[x,\ 0_{D-d}]}{\sqrt{\frac{1}{D}\sum_{i=0}^{d} x_i^{2}}} \odot [\alpha\mu,\ \xi] \tag{31}$$

$$= \left[ \frac{\sqrt{D}}{\sqrt{d}} \frac{x}{\sqrt{\frac{1}{d}\sum_{i=0}^{d} x_i^{2}}} \odot \alpha\mu,\ 0_{D-d} \right] \tag{32}$$

By observing equation 32, we can conclude that, to achieve function preservation, $\alpha = \frac{\sqrt{d}}{\sqrt{D}}$. Finally, we can conclude:

$$RMSNorm'(x') = [RMSNorm(x),\ 0_{D-d}] \tag{33}$$

**Depth**  In depth, preserving only the residual part and masking the MHA and SwiGLU layers can achieve function preservation:

$$Y = X + MHA(RMSNorm(X)) \odot c \tag{34}$$
$$Y = X + SwiGLU(RMSNorm(X)) \odot c \tag{35}$$
$$c = 0_D \rightarrow 1_D \tag{36}$$

## A.6 Details of $G_{\text{learn}}$

Using $G_{\text{learn}}$ for width expansion, for the embedding layer $E \in \mathbb{R}^{V \times d}$, the parameter $B_{emb} \in \mathbb{R}^{D \times d}$ is defined as follows:

$$E' = E B_{emb}^T \tag{37}$$

For Attention layer, where $W_Q, W_K, W_V,$ and $W_O \in \mathbb{R}^{d \times d}$, and RMSNorm $\mu_1 \in \mathbb{R}^d$, the parameters $B_Q, B_K,$ and $B_V \in \mathbb{R}^{D \times d}$, we have:

$$\begin{cases} W_Q' &= B_Q W_Q B_{emb}^T \\ W_K' &= B_K W_K B_{emb}^T \\ W_V' &= B_V W_V B_{emb}^T \\ W_O' &= B_{emb} W_O B_V^T \\ \mu_1' &= B_{emb} \mu_1 \end{cases} \tag{38}$$

For MLP, where $W_{up}, W_{gate} \in \mathbb{R}^{d_{mlp} \times d}$, $W_{down} \in \mathbb{R}^{d \times d_{mlp}}$, RMSNorm $\mu_2 \in \mathbb{R}^d$, the parameter $B_{mlp} \in \mathbb{R}^{D_{mlp} \times d_{mlp}}$, we have:

$$\begin{cases} W_{up}' &= B_{mlp} W_{up} B_{emb}^T \\ W_{down}' &= B_{emb} W_{mlp} B_{mlp}^T \\ W_{gate}' &= B_{mlp} W_{gate} B_{emb}^T \\ \mu_2' &= B_{emb} \mu_2 \end{cases} \tag{39}$$

For the output head $W_{head} \in \mathbb{R}^{V \times d}$, we have:

$$W_{head}' = W_{head} B_{emb} \tag{40}$$

Using $G_{\text{learn}}$ for depth expansion, consider a transformer model with $L_1$ layers, we use $G_{\text{learn}}$ to expand it to $L_2$ layers. For $l \in \{1, 2, \cdots, L_2\}$:

$$\begin{cases} W_l^{Q'} &= \sum_{j=1}^{L_1} D_{l,j}^Q W_j^Q \\ W_l^{K'} &= \sum_{j=1}^{L_1} D_{l,j}^K W_j^K \\ W_l^{V'} &= \sum_{j=1}^{L_1} D_{l,j}^V W_j^V \\ W_l^{O'} &= \sum_{j=1}^{L_1} D_{l,j}^O W_j^O \\ \mu_l^{(ln1)'} &= \sum_{j=1}^{L_1} D_{l,j}^{(ln1)} \mu_j^{(ln1)} \end{cases} \tag{41}$$

where $D^{Q,K,V,O,ln1} \in \mathbb{R}^{L_2 \times L_1}$ represents learnable parameters. These parameters are used to expand the MHA vertically in depth. Similarly, for SwiGLU, we also perform expansion using a similar method. Formally, this can be written as:

$$\begin{cases} W_l^{up'} &= \sum_{j=1}^{L_1} D_{l,j}^{up} W_j^{up} \\ W_l^{down'} &= \sum_{j=1}^{L_1} D_{l,j}^{down} W_j^{down} \\ W_l^{gate'} &= \sum_{j=1}^{L_1} D_{l,j}^{gate} W_j^{gate} \\ \mu_l^{(ln2)'} &= \sum_{j=1}^{L_1} D_{l,j}^{(ln2)} \mu_j^{(ln2)} \end{cases} \tag{42}$$

where $D^{up,down,gate,ln2} \in \mathbb{R}^{L_2 \times L_1}$ represents learnable parameters used for expanding SwiGLU in the depth.

## B    LLMs Framework and Training Details

**Embedding**    Consider a vocabulary size $V$ and embedding size $d$. Then, the embedding matrix $E \in \mathbb{R}^{V \times d}$, and the one-hot vector for input tokens $X$ is denoted as $\mathbb{1}_X \in \mathbb{R}^{T \times V}$, where $T$ is the sequence length. Formally, it can be written as:

$$Embedding(X) = \mathbb{1}_X E \tag{43}$$

for $i, v \in [V]$, where $i \neq j$, it is guaranteed that $E_i \neq E_j$.

**Multi-Head Attention**    Multi-Head Attention (MHA) consists of multiple attention heads, each of which computes its own self-attention. The results of these attention heads are then concatenated and projected to obtain the following output:

$$
\begin{aligned}
Q_i, K_i, V_i &= XW_i^Q, XW_i^K, XW_i^V \\
H_i &= softmax(\frac{Q_i K_i^T}{\sqrt{d_h}})V_i \\
MHA(X) &= Concat(H_1, \cdots, H_n)W^O
\end{aligned}
\tag{44}
$$

here, the input $X \in \mathbb{R}^{T \times d}$, parameters $W_i^Q \in \mathbb{R}^{d \times d_h}$, $W_i^K \in \mathbb{R}^{d \times d_h}$, $W_i^V \in \mathbb{R}^{d \times d_h}$, and $W^O \in \mathbb{R}^{d \times d}$, where $n \times d_h = d$.

**Feed Forward Network**    The Feed Forward Network (FFN) consists of two linear layers and the activation function GeLU. Typically, the two linear layers first perform an up-projection to $d_{FFN}$ and then down-project back to the dimension $d$. Therefore, FFN is defined as:

$$FFN(X) = GeLU(XW_{up})W_{down} \tag{45}$$

where the input $X \in \mathbb{R}^{T \times d}$, parameter $W_{up} \in \mathbb{R}^{d \times d_{FFN}}$ and $W_{down} \in \mathbb{R}^{d_{FFN} \times d}$.

**SwiGLU**    LLaMA replaces the original FFN in the Transformer Decoder with SwiGLU, resulting in improved performance. SwiGLU consists of three linear layers and the swiglu activation function. It can be defined as:

$$SwiGLU(X) = (XW_{gate} \odot swiglu(XW_{up}))W_{down} \tag{46}$$

where $\odot$ means the element-wise multiplication, the input $X \in \mathbb{R}^{T \times d}$, parameter $W_{up} \in \mathbb{R}^{d \times d_{FFN}}$, $W_{gate} \in \mathbb{R}^{d \times d_{FFN}}$ and $W_{down} \in \mathbb{R}^{d_{FFN} \times d}$.

**RMSNorm**    Before MHA, FFN, or SwiGLU, there is a layer of RMSNorm to enhance the stability of the model. Compared to LayerNorm, RMSNorm is simpler in form. Formally, it can be written as:

$$RMSNorm(X) = \frac{X}{\sqrt{\frac{1}{d} \sum_{i=1}^{d} X_i^2}} \odot \mu \tag{47}$$

where $X \in \mathbb{R}^{T \times d}$, parameter $\mu \in \mathbb{R}^d$.

## B.1 LLMs Training with Growth Operator

---

**Algorithm 2** LLMs Training with Growth Operator

---

**Input:** Growth operator $G$, Loss function $\mathcal{L}$, Iterative optimizer $A$. Dataset $\{d_1, d_2, \cdots, d_k\}$ for base model. Dataset $\{D_1, D_2, \cdots, D_K\}$ for target model.
**Output:** Target Model $\mathcal{M}_K$
**Initial Phase**: Initialize a base model $M_0$ from scratch.
**for** $t = 1 \ to \ k$ **do**                                      ▷ Base Model Training
   $loss = \mathcal{L}(M_{t-1}, d_t)$
   $M_t \leftarrow \mathcal{A}(M_{t-1}, loss)$
**end**
$\mathcal{M}_0 = G(M_k)$
**for** $t = 1 \ to \ K$ **do**                                      ▷ Target Model Training
   $loss = \mathcal{L}(\mathcal{M}_{t-1}, D_t)$
   $\mathcal{M}_t \leftarrow \mathcal{A}(\mathcal{M}_{t-1}, loss)$
**end**

---

## B.2 Details of Speedup Calculation

We calculate speedup $sp$ between operator $G$ and $scratch$ model pre-training by:

$$sp = \frac{FLOPs_{scratch}}{FLOPs_G} - 1 \tag{48}$$

where $FLOPs_{scratch}$ and $FLOPs_G$ represent the FLOPs required by the scratch model and the $G$ model, respectively, to achieve the same loss.

## B.3 Details of Training Settings

We use TinyLlama [7] [45] as our pre-training codebase. We employ FSDP (Fully Sharded DataParallel) along with FlashAttention [46] 2.0, and other acceleration techniques. We use the open-source dataset Slimpajama-627B [8] [47] for pre-training. The hyperparameters used for each model size are listed in Table 1. Our 7B model is trained over around 100B tokens per day on an NVIDIA Hopper cluster.

Table 1: Hyperparameters

| Size | Context Length | Batch Size | max-LR | min-LR | Warmup Steps | LR Scheduler |
|------|----------------|------------|--------|--------|--------------|--------------|
| **410M** | 2048 | 2M tokens | 6e-4 | 6e-5 | 3000 | cosine |
| **1.1B** | 2048 | 2M tokens | 3e-4 | 3e-5 | 3000 | cosine |
| **3B** | 2048 | 2M tokens | 1.6e-4 | 1.6e-5 | 3000 | cosine |
| **7B** | 2048 | 2M tokens | 1e-4 | 1e-5 | 3000 | cosine |

# C Training Loss and Evaluation Results of Four Operators in both Depth and Width growth

We have two small (base) models, one trained with token count $d = 10B$ and another trained with token count $d = 50B$.

---

[7]Apache-2.0 license

[8]The license of Slimpajama-627B includes: Common Crawl Foundation Terms of Use; C4 license; GitHub was limited to MIT, BSD, or Apache licenses only; Books: the_pile_books3 license and pg19 license; ArXiv Terms of Use; Wikipedia License; StackExchange license on the Internet Archive

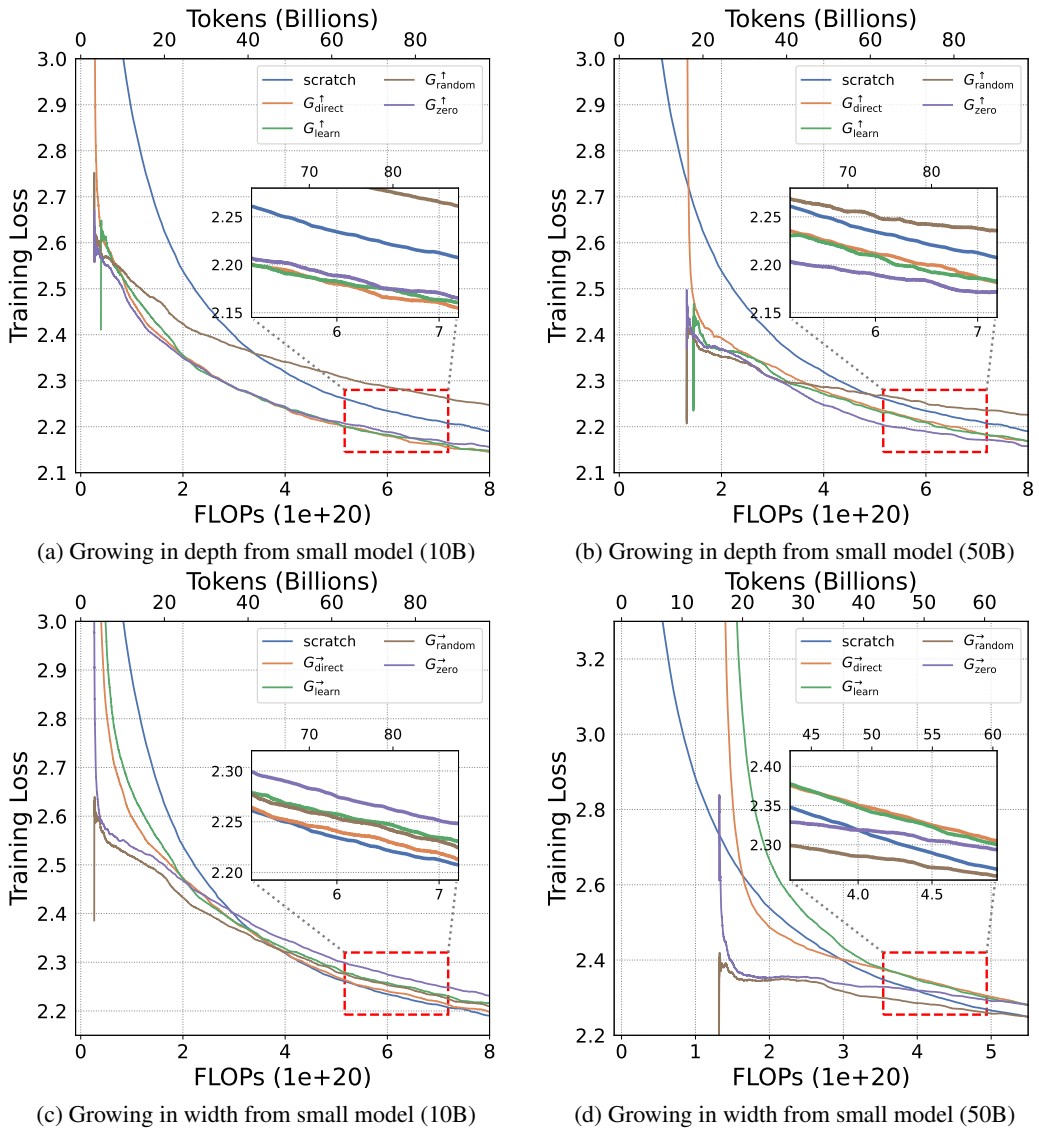

(a) Growing in depth from small model (10B)

(b) Growing in depth from small model (50B)

(c) Growing in width from small model (10B)

(d) Growing in width from small model (50B)

Figure 11: Training Loss on Slimpajama.

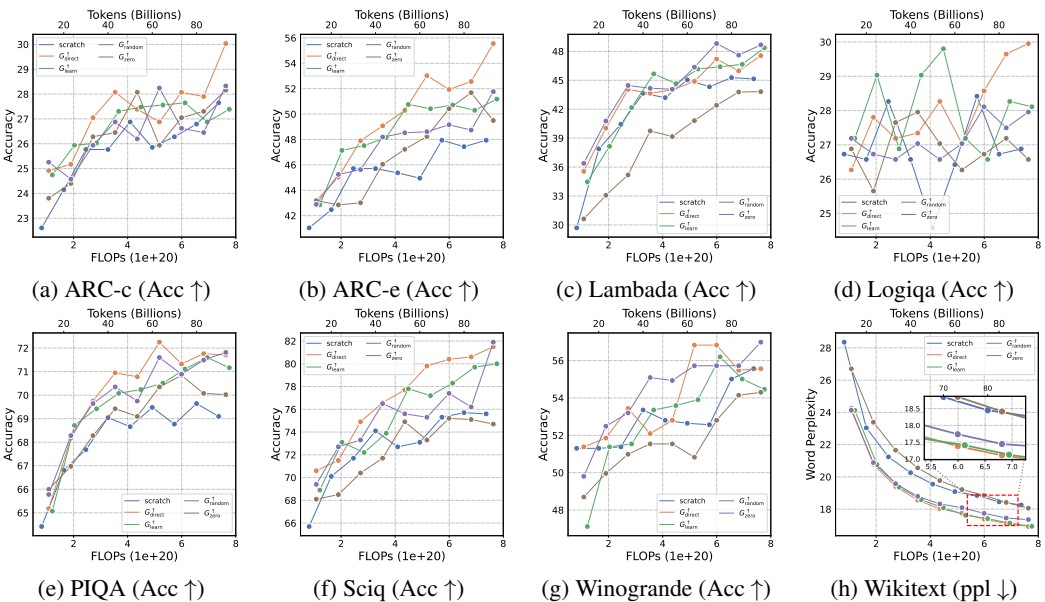

Figure 12: Evaluation results on growth in depth from small model (10B) by four operators.

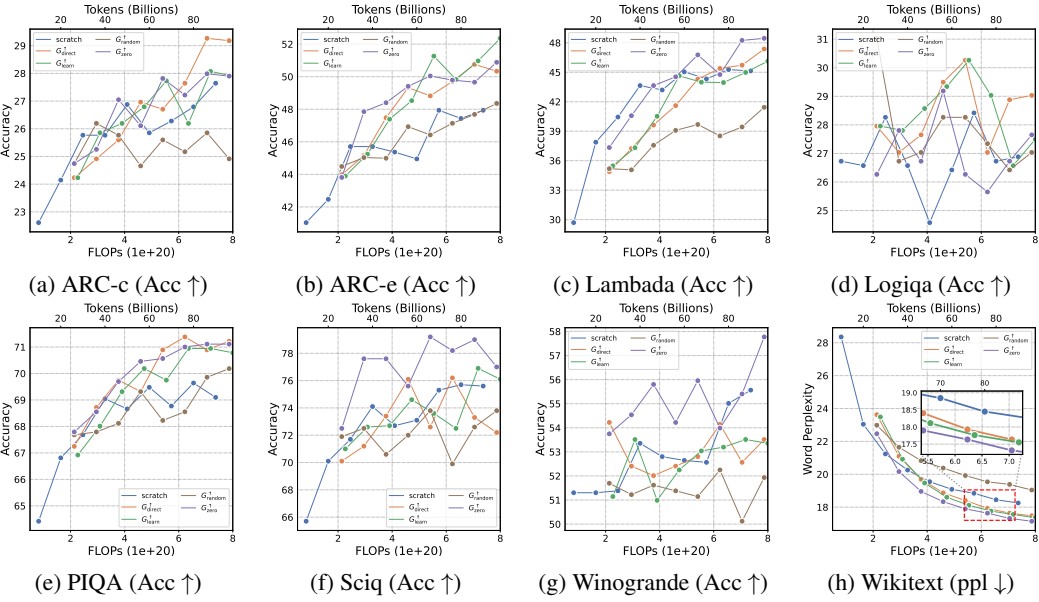

Figure 13: Evaluation results on growth in depth from small model (50B) by four operators.

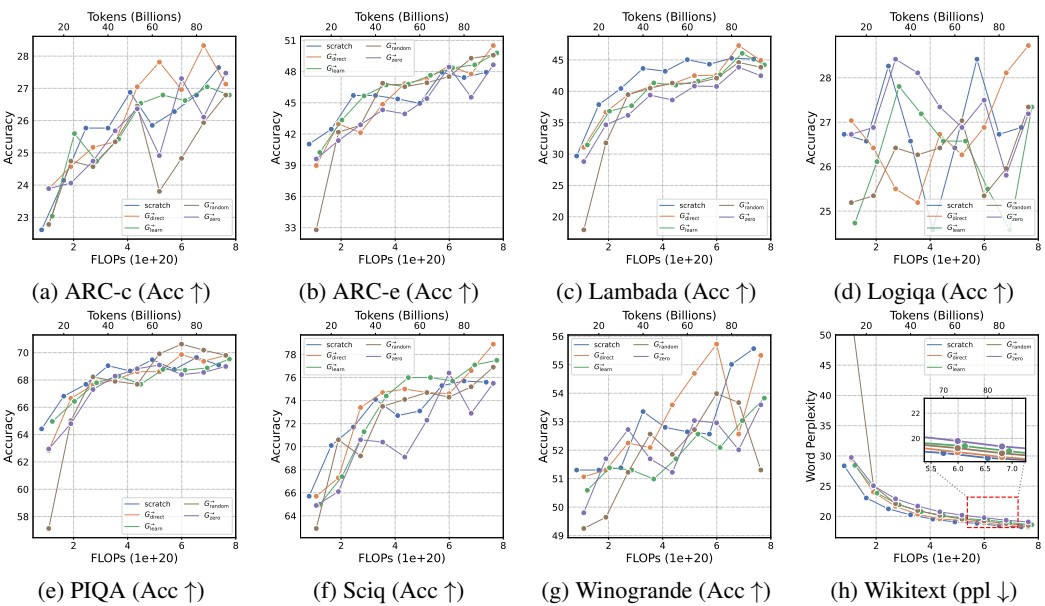

Figure 14: Evaluation results on growth in width from small model (10B) by four operators.

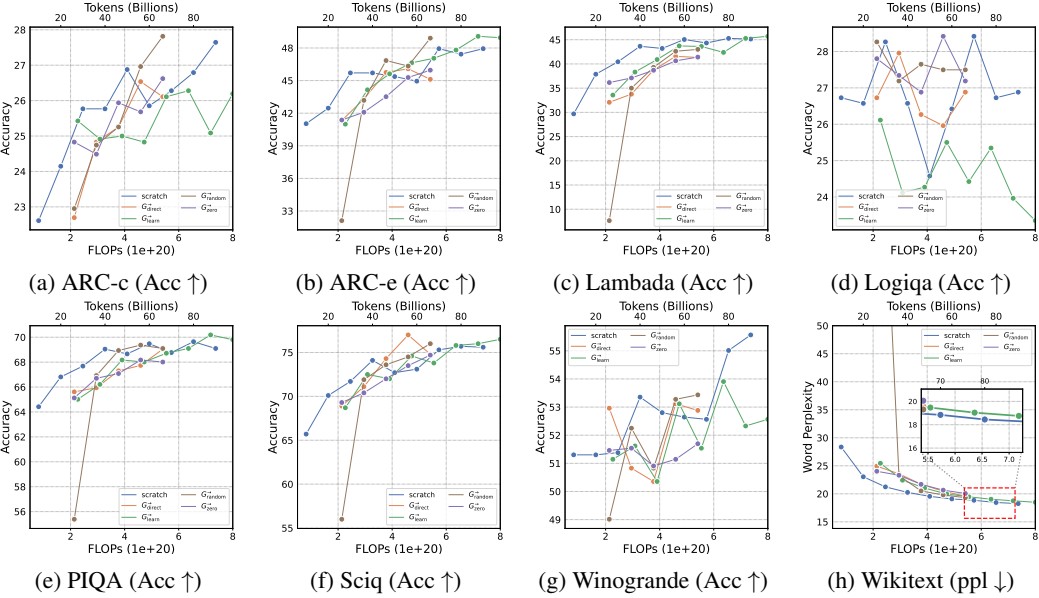

Figure 15: Evaluation results on growth in width from small model (50B) by four operators.

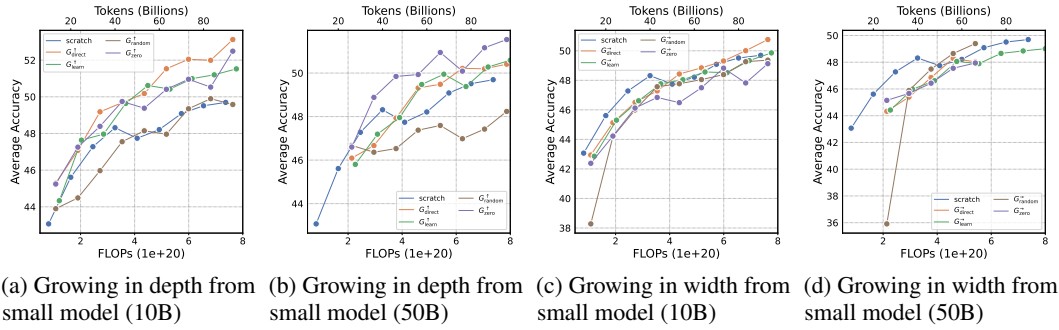

(a) Growing in depth from small model (10B)

(b) Growing in depth from small model (50B)

(c) Growing in width from small model (10B)

(d) Growing in width from small model (50B)

Figure 16: Average accuracy of seven standard NLP benchmarks.

# D   Evaluation Results of Scaling $G_{\text{stack}}$

## D.1   3B

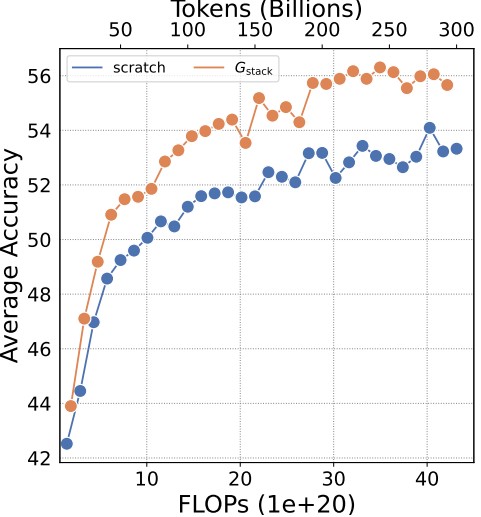

Figure 17: Average accuracy of standard NLP benchmarks at 3B size.

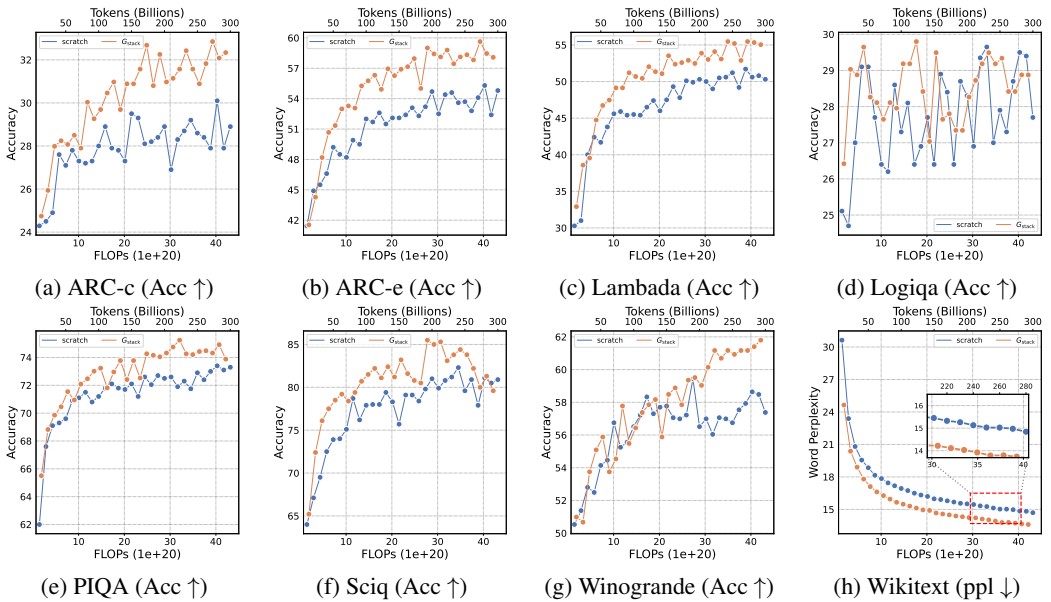

Figure 18: Evaluation results on scratch model and $G_{stack}$ model at 3B size.

## D.2 7B

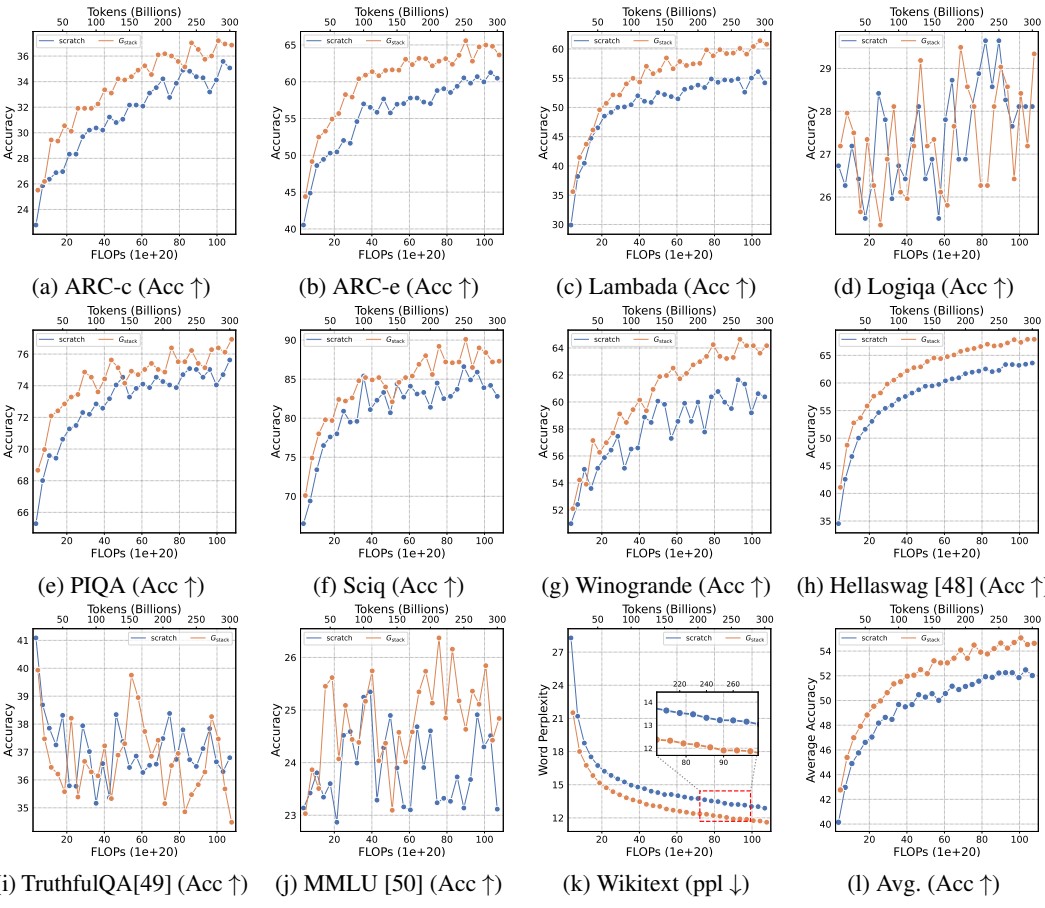

Figure 19: Evaluation results on scratch model and $G_{stack}$ model at 7B size.

## D.3 410M

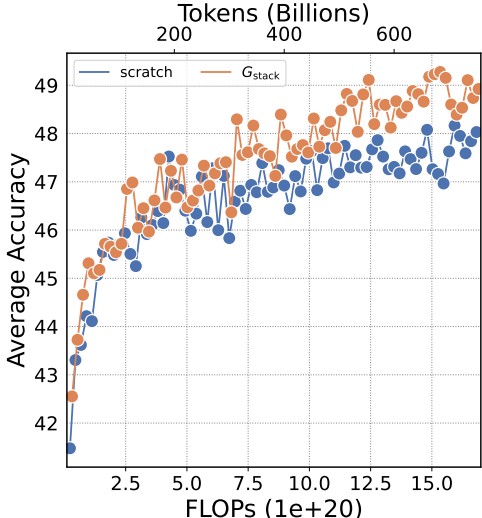

Figure 20: Average accuracy of standard NLP benchmarks at 410M size.

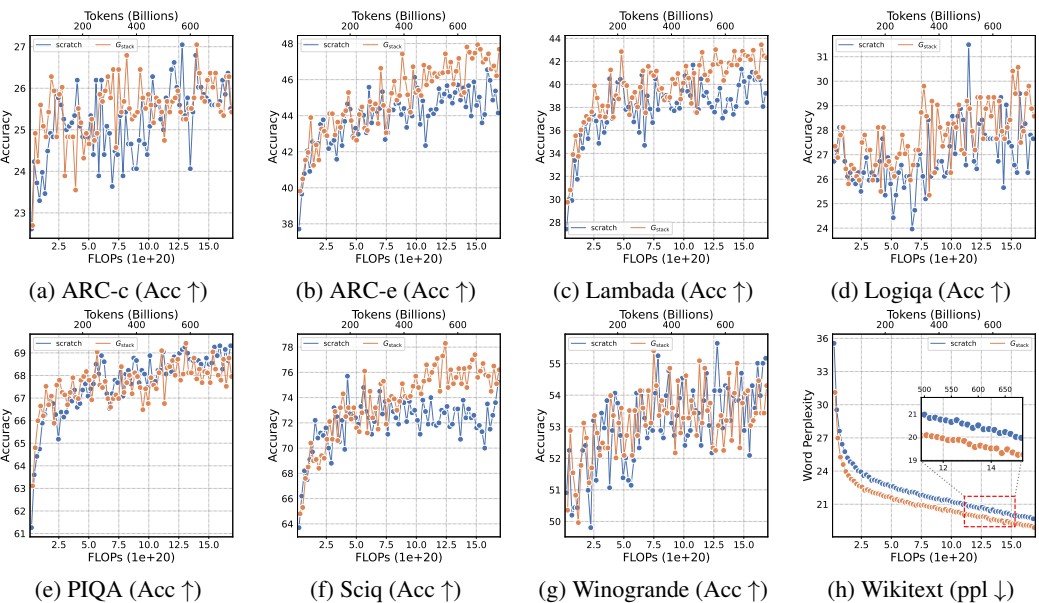

(a) ARC-c (Acc ↑)  (b) ARC-e (Acc ↑)  (c) Lambada (Acc ↑)  (d) Logiqa (Acc ↑)

(e) PIQA (Acc ↑)  (f) Sciq (Acc ↑)  (g) Winogrande (Acc ↑)  (h) Wikitext (ppl ↓)

Figure 21: Evaluation results on scratch model and $G_{\text{stack}}$ model at 410M size.

## D.4 Instruction Tuning Results on 3B

Table 2: Evaluation Results after Instruction-Tuning (Higher better)

| Method | Tokens | Tuning | lambda | arc-c | arc-e | logiqa | piqa | sciq | winogrande | avg |
|--------|--------|--------|--------|-------|-------|--------|------|------|------------|-----|
| scratch | 400B | ✗ | 54.07 | 28.84 | 55.35 | 26.88 | 73.94 | 82.0 | 59.43 | 54.36 |
| | | ✓ | 60.35 | 31.48 | 56.1 | 27.04 | 74.32 | 81.2 | 60.14 | **55.8** |
| $G_{\text{stack}}$ | 290B | ✗ | 55.04 | 32.34 | 58.08 | 28.88 | 73.88 | 79.6 | 61.8 | 55.66 |
| | | ✓ | 61.34 | 34.98 | 59.97 | 29.65 | 75.14 | 80.1 | 60.22 | **57.34** |

# E    Compare with Other Opensource LLMs

In Table 3, we compare the harness evaluation results after training the $G_{stack}$ model and the scratch model (Baseline) for 100B tokens with Pythia-1B [51] and TinyLlama-1.1B, which are trained on the same number of tokens. The comparative results indicate that our baseline performs normally, comparable to pythia-1B. Meanwhile, the $G_{stack}$ model significantly outperforms both the baseline and pythia-1B, demonstrating the acceleration effect of $G_{stack}$ on the pre-training process.

Table 3: Compare with opensource LLMs on 1B

|  | **Pythia-1B** | **TinyLlama-1.1B** | $G_{stack}$**-1.1B** | **Baseline-1.1B** |
|---|---|---|---|---|
| **Datasets** | Pile-300B [52] | Slimpajama-627B& Starcoder | Slimpajama-627B | Slimpajama-627B |
| **Tokens** | 100B | 103B | 100B | 100B |
| **lambada** | **53.52** | - | 48.20 | 47.87 |
| **ARC-c** | 25.59 | 24.32 | **29.18** | 27.21 |
| **ARC-e** | 47.26 | 44.91 | **54.25** | 48.86 |
| **piqa** | 69.31 | 67.30 | **71.98** | 69.64 |
| **logiqa** | **29.49** | - | 28.87 | 25.96 |
| **sciq** | 77.3 | - | **81.1** | 76.8 |
| **winogrande** | 51.22 | 53.28 | **56.03** | 54.53 |
| **Avg.** | 50.53 | - | **52.80** | 50.09 |

# F    Fitting Results for the Growth Factor $g$

Although due to computational resource limitations, we only explore predicting $g$ given $N$ and $C$ on the 1.1B and 3B models, we still attempted to fit using equation:

$$\log_{10}(g) = a \log_{10}(N) + \frac{b}{\log_{10}(C)} + c \tag{49}$$

In the equation 49, $N$ represents the number of target parameters, $g$ represents the growth factor. The fitting result is as follows:

$$\log_{10}(g) = 1.01 \log_{10}(N) - \frac{29.88}{\log_{10}(C)} - 7.36 \tag{50}$$

We also visualize the fitted curves in Figure 22, but the results were mediocre due to the lack of data.

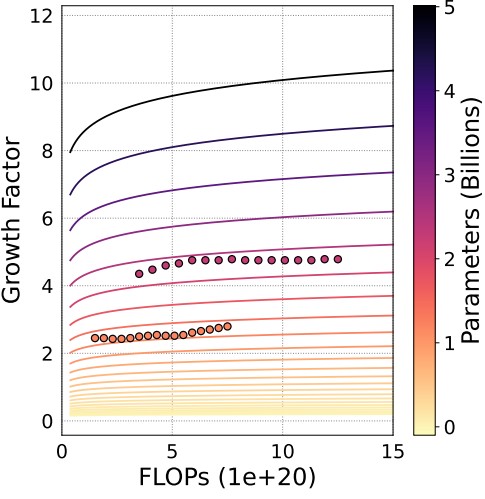

Figure 22: Visualization of the Equation 50.

We give an example of empirical usage of $G_{\text{stack}}$ by using the configurations of Llama2 and Llama3 families [21, 7] to show the estimated optimal base model training tokens $d$ and growth factor $g$ in Table 4.

Table 4: "Stacking Law" Guidelines

| Model | N | D | $d$ | $g$ |
|---|---|---|---|---|
| Llama3-8B | 8B | 15T | 6.58B | 4 |
| Llama2-7B | 7B | 2T | 11.11B | 4 |
| Llama2-13B | 13B | 2T | 15.84B | 4 |
| Llama2-70B | 70B | 2T | 42.48B | 4 |

# G    Training Loss and Evaluation Results of "growth timing" and "growth factor"

## G.1    "Growth Timing" $d$

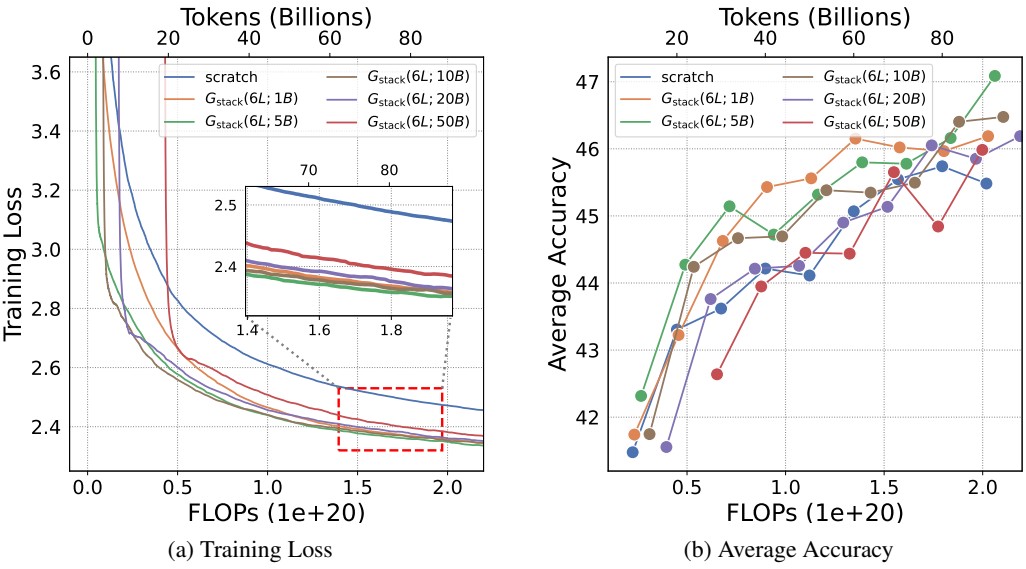

(a) Training Loss

(b) Average Accuracy

Figure 23: Training loss and standard NLP benchmarks average accuracy of 410M.

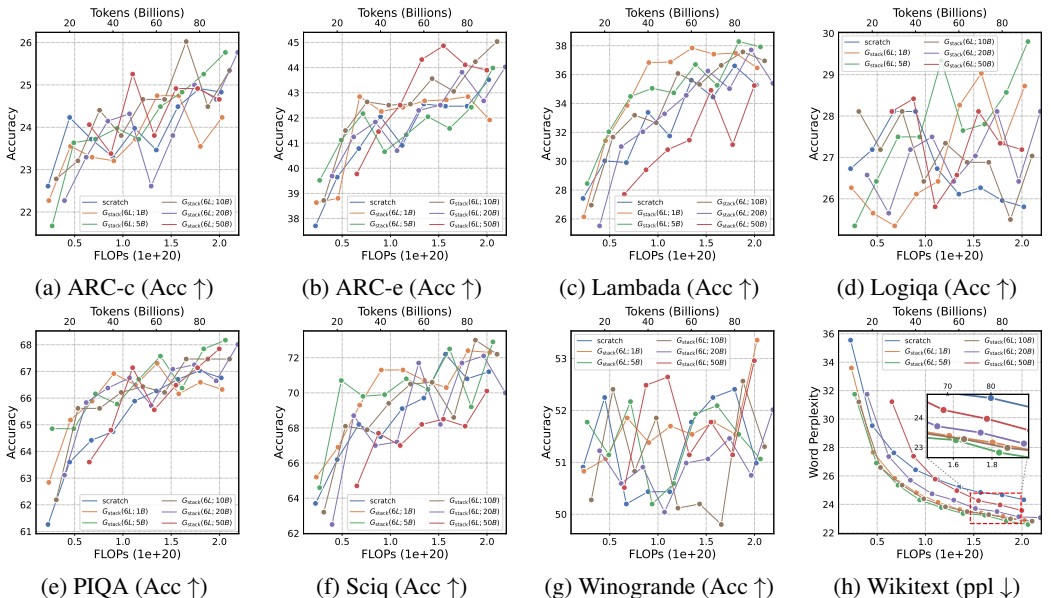

Figure 24: Evaluation results on 410M.

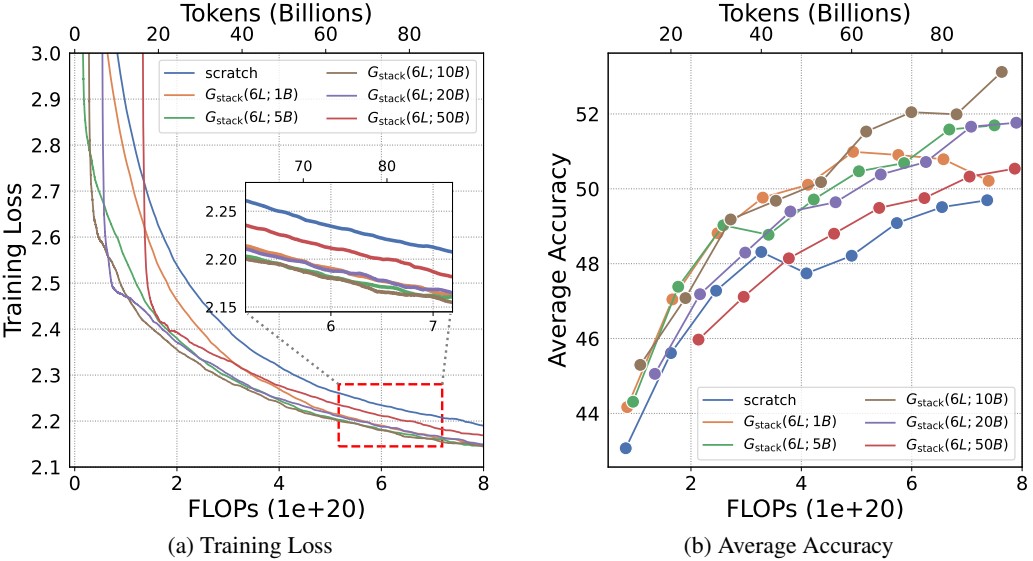

(a) Training Loss

(b) Average Accuracy

Figure 25: Training loss and standard NLP benchmarks average accuracy of 1.1B.

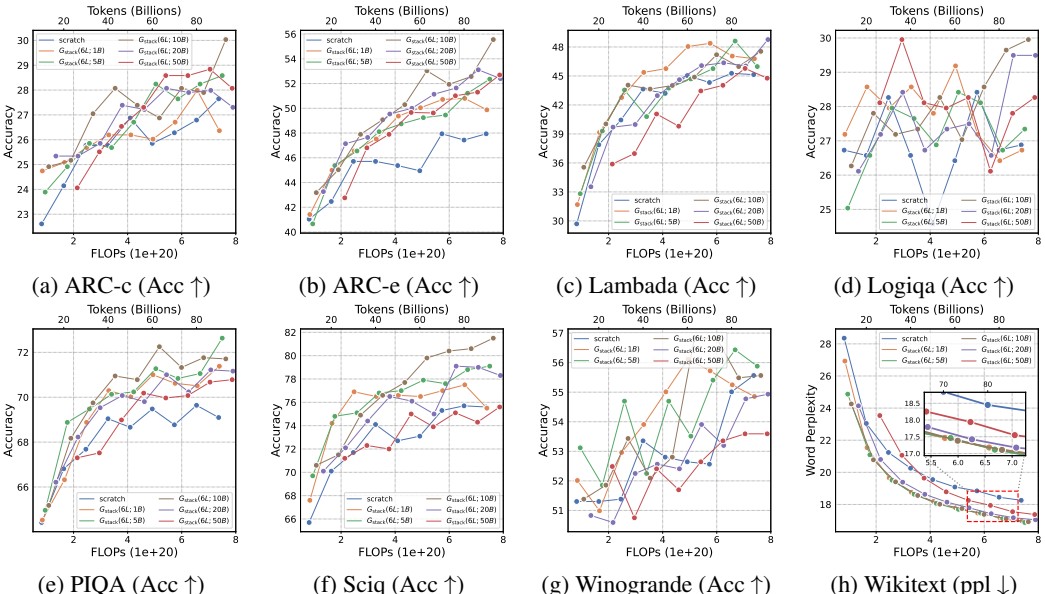

Figure 26: Evaluation results on 1.1B.

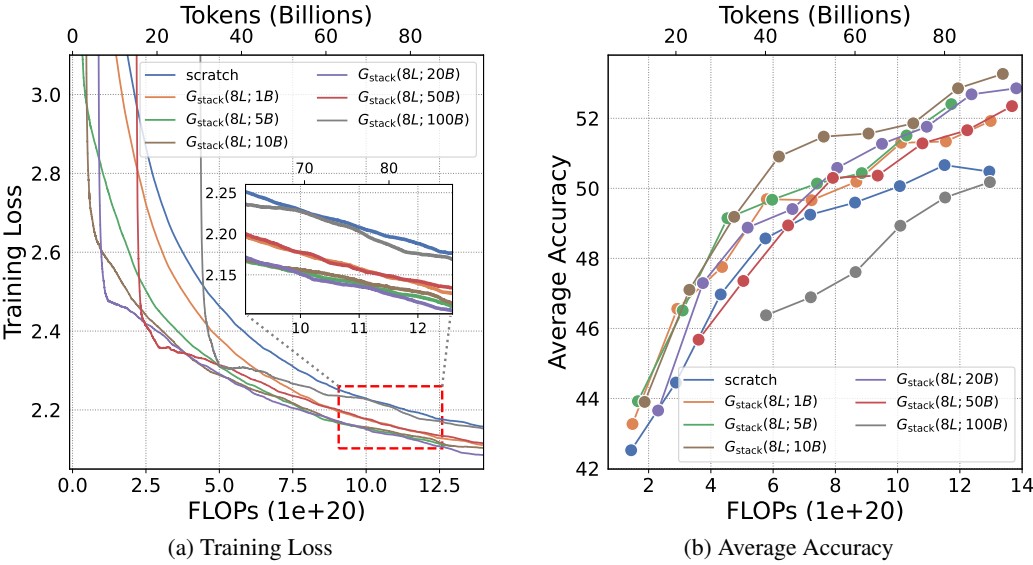

Figure 27: Training loss and standard NLP benchmarks average accuracy of 3B.

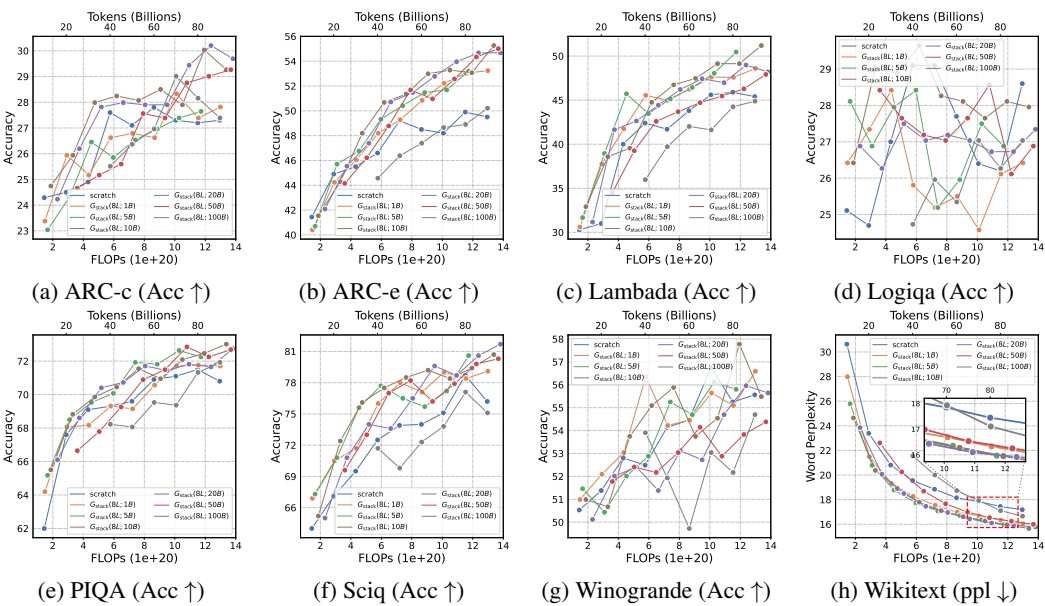

(a) ARC-c (Acc ↑)  (b) ARC-e (Acc ↑)  (c) Lambada (Acc ↑)  (d) Logiqa (Acc ↑)

(e) PIQA (Acc ↑)  (f) Sciq (Acc ↑)  (g) Winogrande (Acc ↑)  (h) Wikitext (ppl ↓)

Figure 28: Evaluation results on 3B.

## G.2  "Growth Factor" $g$

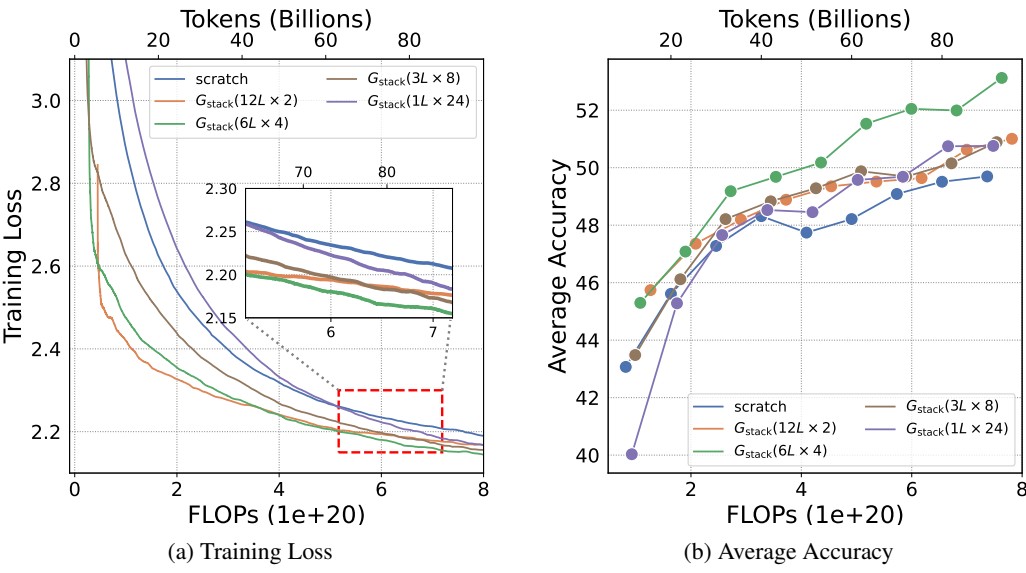

(a) Training Loss  (b) Average Accuracy

Figure 29: Training loss and standard NLP benchmarks average accuracy of 1.1B.

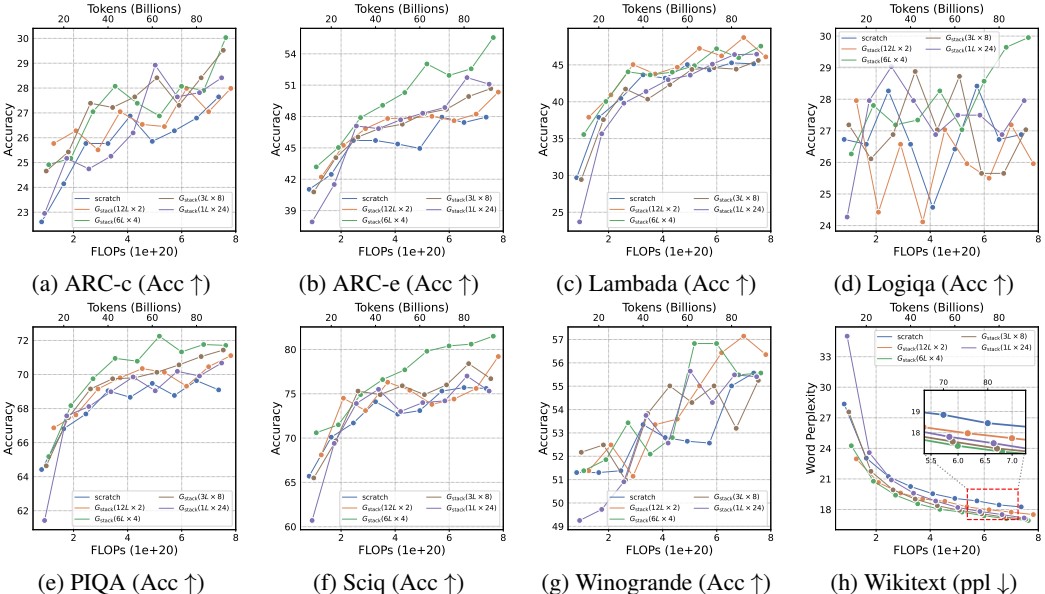

(a) ARC-c (Acc ↑)  (b) ARC-e (Acc ↑)  (c) Lambada (Acc ↑)  (d) Logiqa (Acc ↑)

(e) PIQA (Acc ↑)  (f) Sciq (Acc ↑)  (g) Winogrande (Acc ↑)  (h) Wikitext (ppl ↓)

Figure 30: Evaluation results on 1.1B.

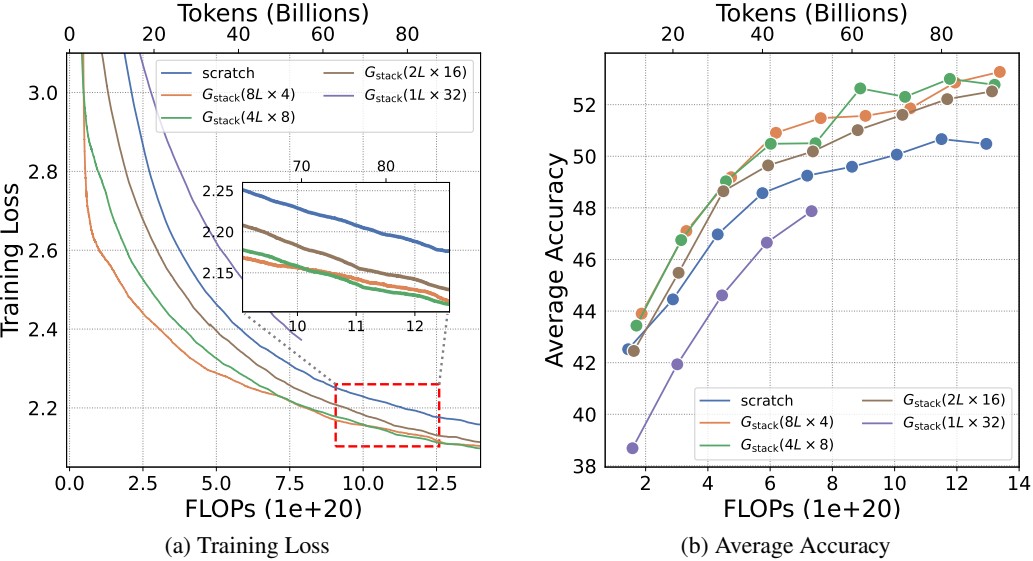

(a) Training Loss          (b) Average Accuracy

Figure 31: Training loss and standard NLP benchmarks average accuracy of 3B.

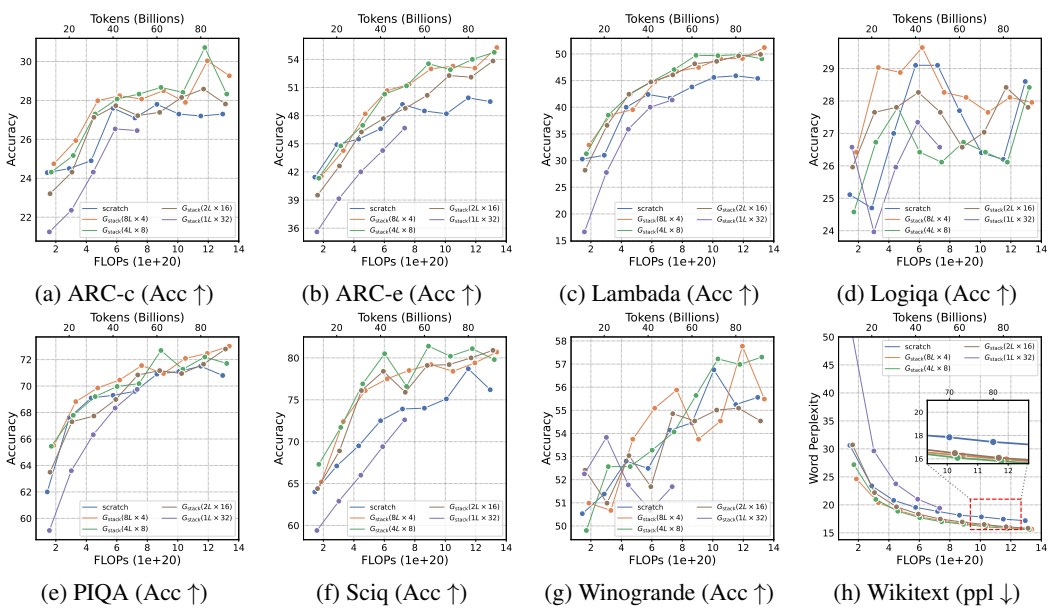

Figure 32: Evaluation results on 3B.

# H   Discussion on "How to stack?" and Evaluation Results

## H.1   Training Loss and Evaluation Results of Gradual Stack

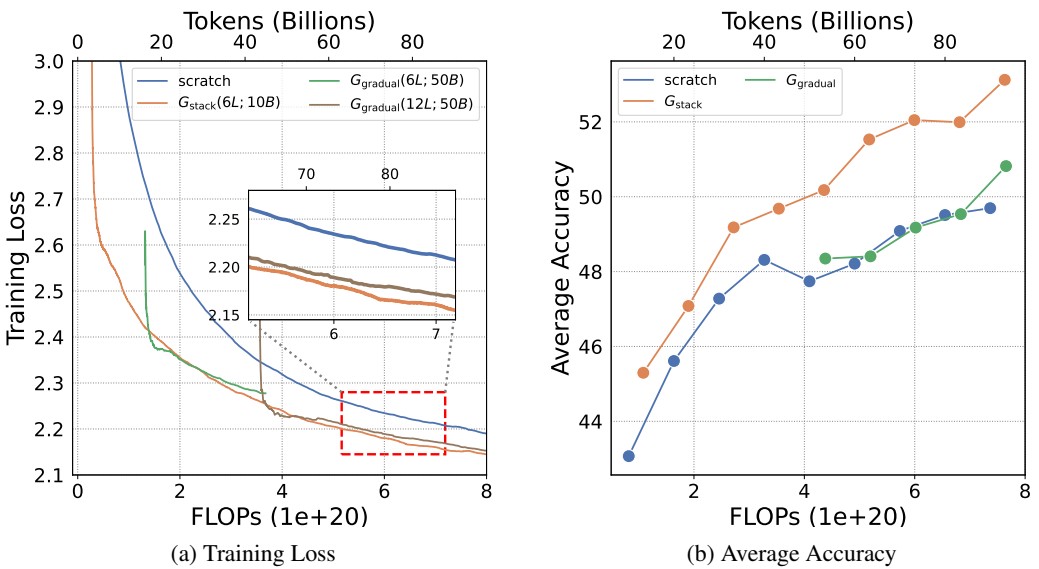

Figure 33: Training loss and standard NLP benchmarks average accuracy of scratch, $G_{\text{stack}}$ and $G_{gradual}$.

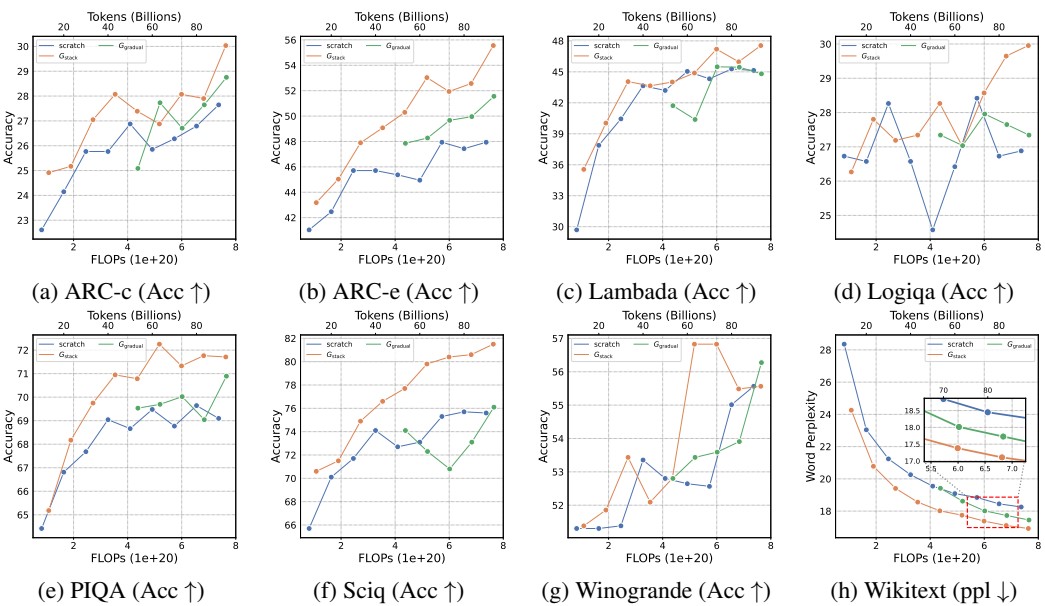

Figure 34: Evaluation results on scratch, $G_{\text{stack}}$ and gradual stacking in StackBert.

## H.2 Ablation: $f_2 \circ f_1 \circ f_0 \circ f_2 \circ f_1 \circ f_0$ or $f_2 \circ f_2 \circ f_1 \circ f_1 \circ f_0 \circ f_0$ (interpolation)

To investigate whether the connections between layers affect the performance of stacking, we conduct a comparison of two approaches for stacking small models into larger ones. We explore two approaches for stacking small models into larger ones. The first approach involves taking the entire small model as a unit and directly stacking it, which can retain the connections between most layers. The second approach involves replicating and interleaving each layer in the small model, which almost break the connections. To measure the degree of retention of inter-layer connections after stacking, we define the connection rate $R_c$:

$$R_c = \frac{Con_r}{Con_{all}} \tag{51}$$

where the $Con_r$ is number of retained connections, the $Con_{all}$ is number of all connections.

For example, if we had a small model with three layers, denoted as $f_2 \circ f_1 \circ f_0$, and desired a model depth of 6, the first approach would result in $f_2 \circ f_1 \circ f_0 \circ f_2 \circ f_1 \circ f_0$, where its $R_c = 80\%$. The second approach would result in $f_2 \circ f_2 \circ f_1 \circ f_1 \circ f_0 \circ f_0$, where its $R_c = 40\%$.

In our experiments, we stack a small model with 8 layers to a 24 layers target model. The growth timing $d$ is $10B$ tokens and growing factor $s$ is 3. The $R_c$ of $G_{\text{stack}}$ is $91.3\%$ and the $R_c$ of $G_{interpolate}$ is $30.4\%$. We report the training loss and standard NLP benchmarks average accuracy in Figure 35. At the beginning of training, interpolated stacking perform as well as stacking entire small model. However, as the training continues, the performance of interpolated stacking deteriorates.

Therefore, we can conclude that the higher the connection rate of stacking, the better the effect of stacking. In Appendix H.3, we continue to validate this conclusion.

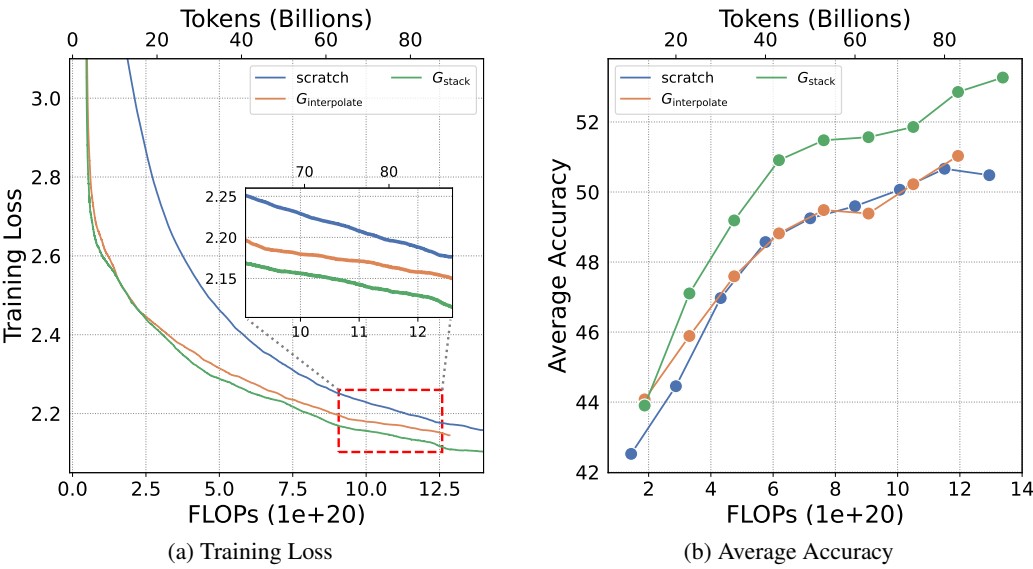

(a) Training Loss

(b) Average Accuracy

Figure 35: Training loss and standard NLP benchmarks average accuracy of scratch, $G_{\text{stack}}$ and interpolation.

We also report the details of evaluation results about 8 standard NLP benchmarks.

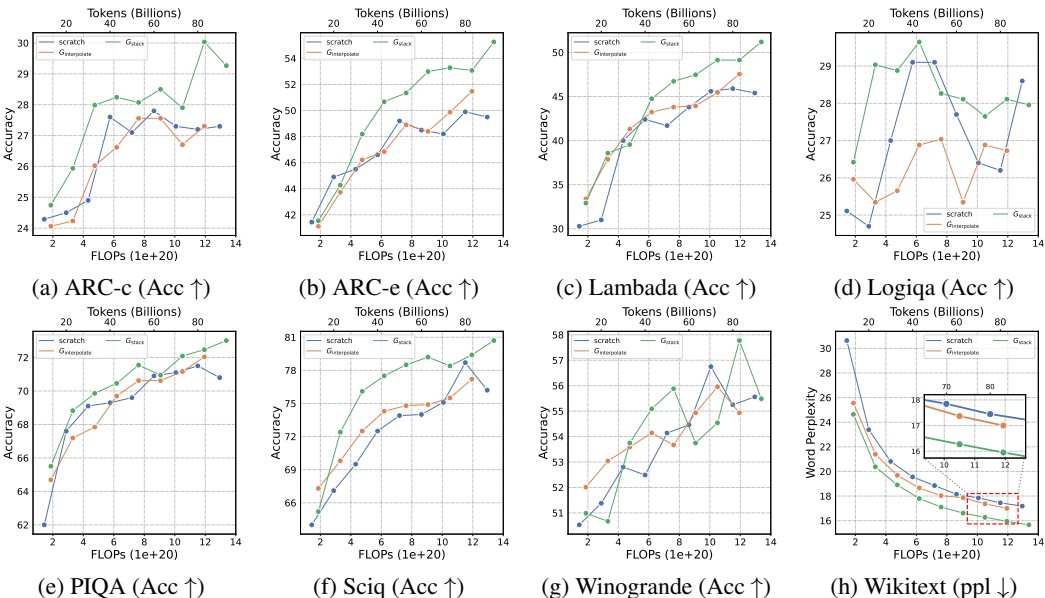

(a) ARC-c (Acc ↑)  (b) ARC-e (Acc ↑)  (c) Lambada (Acc ↑)  (d) Logiqa (Acc ↑)

(e) PIQA (Acc ↑)  (f) Sciq (Acc ↑)  (g) Winogrande (Acc ↑)  (h) Wikitext (ppl ↓)

Figure 36: Evaluation results on scratch, $G_{\text{stack}}$ and interpolation.

## H.3 Ablation: Partial Stacking

Partial stacking has been explored in LLMs like LlamaPro [42], Solar [43]. But their goal is to stack an off-the-shelf LLMs such as Llama2, while our aim is to accelerate LLM pre-training process.

To explore stacking which layers of the small model can achieve the best performance, we conduct experiments on partial stacking. In our experiments, we stack a small model with 6 layers ($\{L_1, L_2, \cdots, L_6\}$) to a 24 layers target model. We set growth timing $d = 10B$ tokens and growth factor $g = 4$. For simplicity, we use a format such as 1-234*7-56 to denote stacking 234 layers 7 times.

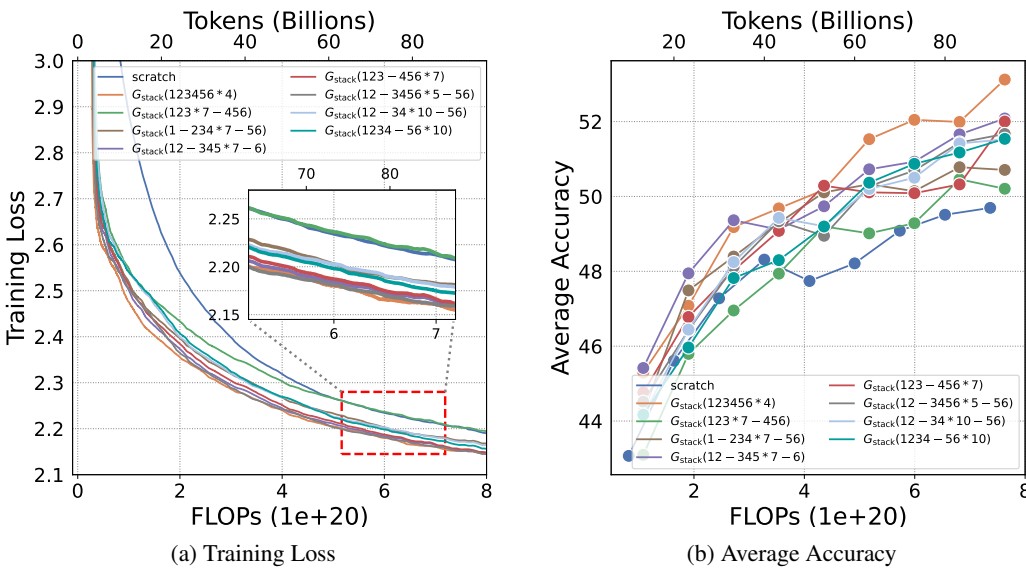

|  | |
|---|---|
| (a) Training Loss | (b) Average Accuracy |

Figure 37: Training loss and standard NLP benchmarks average accuracy of scratch, $G_{\text{stack}}$ and other partial stacking.

We report the training loss and standard NLP benchmarks average accuracy in Figure 37. By observing the loss curves in Figure 37a, we can find that the eight partial stacking methods are clearly divided into three groups based on their loss. The first group, {123456*4, 12-3456*5-56, 12-345*7-6, 123-456*7}, achieves the best performance. The second group consisting of {1234-56*10, 12-34*10-56, 1-234*7-56}, performs just so-so. The third group, {123*7-456}, performs poorly, even worse than the baseline.

In Table 5, we summarize the eight partial stacking and calculate the $R_c$ of each partial stacking methods based on Equation 51.

For partial stacking, we conclude that: all > middle ≈ back ≫ front. Meanwhile, when the stacked parts are the same, the larger the $R_c$, the better the performance.

Table 5: $R_c$ and stacked parts of each partial stacking method

| Group | Method | Stacked parts | $R_c$ |
|---|---|---|---|
| **First** | 123456*4 | all | 87.0% |
| | 12-3456*5-56 | middle-back | 78.3% |
| | 12-345*7-6 | middle-back | 74.0% |
| | 123-456*7 | back | 74.0% |
| **Second** | 1234-56*10 | back | 60.7% |
| | 12-34*10-56 | middle | 60.7% |
| | 1-234*7-56 | front-middle | 74.0% |
| **Third** | 123*7-456 | front | 74.0% |

Then, we report the evaluation results here.

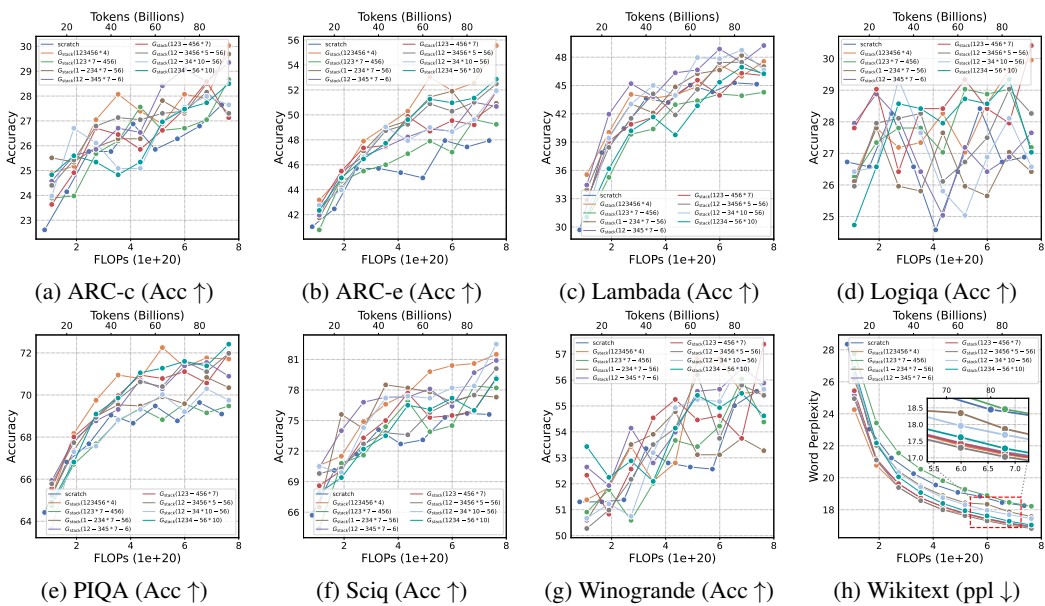

Figure 38: Evaluation results on scratch, $G_{\text{stack}}$ and other partial stacking.

## H.4 Compare with Pythia, OLMo and Amber on 7B Size

Table 6: Compare with opensource 7B LLMs on 130B tokens.

| Datasets | **Pythia-6.9B** Pile-300B [52] | **OLMo-7B** [53] Dolma [55] | **Amber-7B** [54] Amber | $G_{\text{stack}}$**-7B** Slimpajama-627B |
|---|---|---|---|---|
| **Tokens** | 130B | 133B | 132B | 130B |
| **ARC-c** | 33.28 | 28.58 | 29.01 | **35.24** |
| **ARC-e** | 59.81 | 51.60 | 55.05 | **63.64** |
| **boolq** | 63.39 | 55.05 | 60.18 | **66.45** |
| **hellaswag** | 60.03 | 54.52 | 61.21 | **65.85** |
| **lambada** | **65.11** | 49.91 | 57.13 | 57.93 |
| **logiqa** | **28.88** | 28.42 | 26.73 | 26.88 |
| **obqa** | 37.20 | 33.60 | **37.40** | 36.40 |
| **piqa** | 75.03 | 74.43 | 76.01 | **76.82** |
| **sciq** | 82.7 | 74.4 | 82.0 | **85.9** |
| **winogrande** | 60.14 | 53.75 | 56.83 | **62.75** |
| **Avg.** | 56.56 | 50.43 | 54.16 | **57.79** |
| **Wikitext** | 13.3340 | 18.4690 | 15.6202 | **12.5635** |

# I Details of Function Preserving

## I.1 Function Preserving

Function preservation is a key concept that underlies diverse model growth approaches. It entails ensuring consistent output from a model, regardless of its expansion. Mathematically, let us define a function as $F$ and a growth operator as $G$. The ultimate aim is to apply the operator $G$ to the function $F$, thereby obtaining the target function denoted as $\mathcal{F}$. The core objective here is to maintain the model's function to generate the same output for a given input. Formally,

$$\forall x, \mathcal{F}(x) = F(x), \text{ where } \mathcal{F} = G(F) \tag{52}$$

## I.2 Breaking Function Preserving by Adding Noise

For the down projection in SwiGLU and the output projection in MultiHeadAttention, we apply noise:

$$W_{noise} \leftarrow (1-\alpha)W + \alpha\epsilon \quad \text{where } \epsilon \sim \mathcal{N}(0, \frac{1}{d \times l^2}) \tag{53}$$

For the Embedding Layer and other Linear Layers, we apply noise:

$$W_{noise} \leftarrow (1-\alpha)W + \alpha\epsilon \quad \text{where } \epsilon \sim \mathcal{N}(0, \frac{2}{5d}) \tag{54}$$

**Adding Noise on $G_{\text{direct}}$ to Break FP**

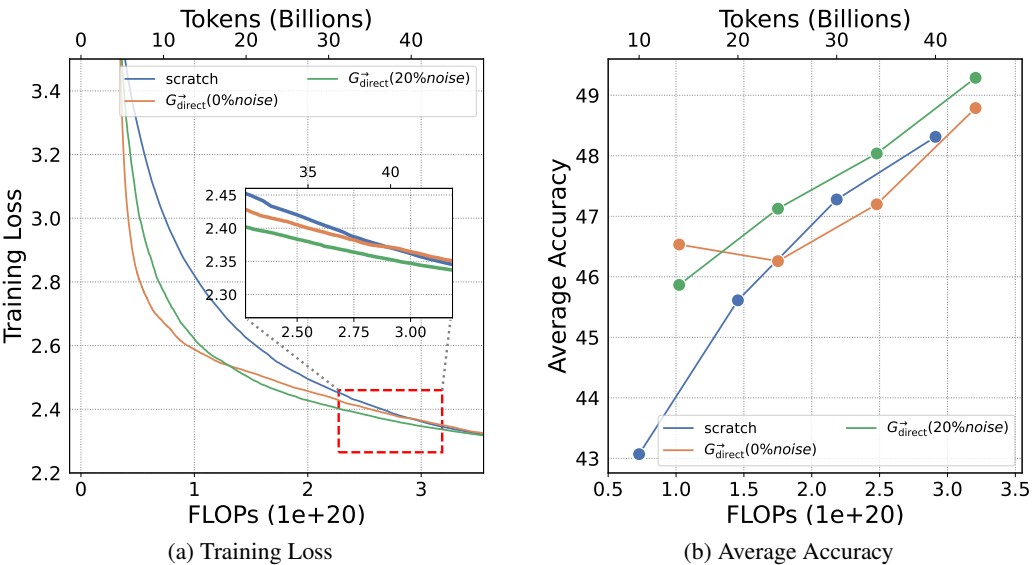

(a) Training Loss        (b) Average Accuracy

Figure 39: Training loss and standard NLP benchmarks average accuracy of scratch, $G_{\overrightarrow{\text{direct}}}$ and $G_{\overrightarrow{\text{direct}}}$ with 20% noise.

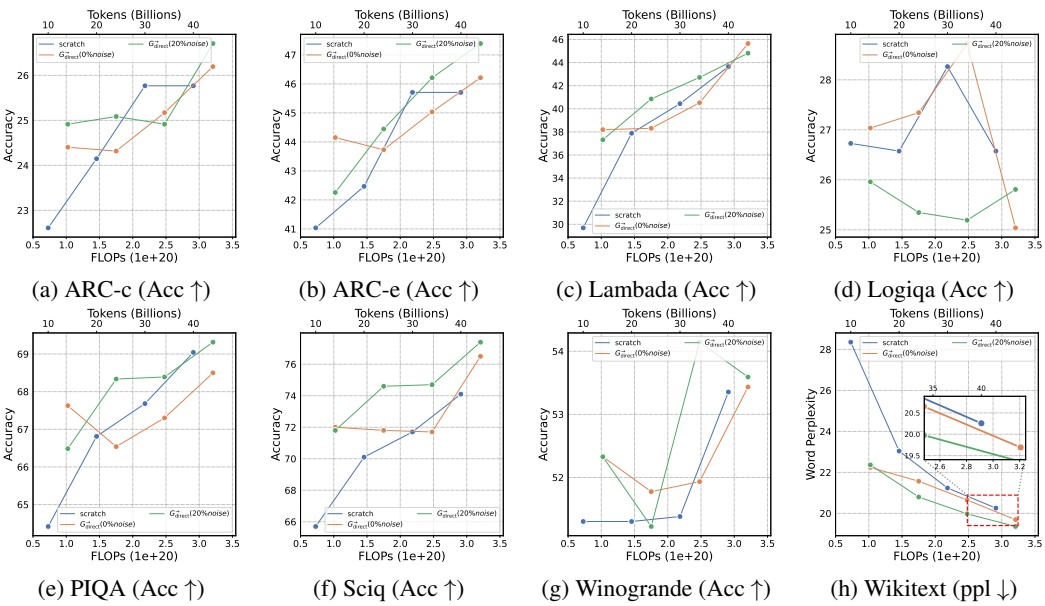

Figure 40: Evaluation results on scratch, $G^{\rightarrow}_{\text{direct}}$ and $G^{\rightarrow}_{\text{direct}}$ with 20% noise.

## Training Loss And Evaluation Results on Adding Noise $G^{\rightarrow}_{\textbf{direct}}$

**Adding Noise on $G_{\textbf{stack}}$**    Since adding noise actually improve the $G_{\text{direct}}$ performance, we also add noise on $G_{\text{stack}}$.

We stack an 8 layers small model to 24 layers, and then add noise with $\alpha = 0.2$. We report training loss and standard NLP benchmarks average accuracy in Figure 41. Adding noise demonstrates an advantage in Training loss.

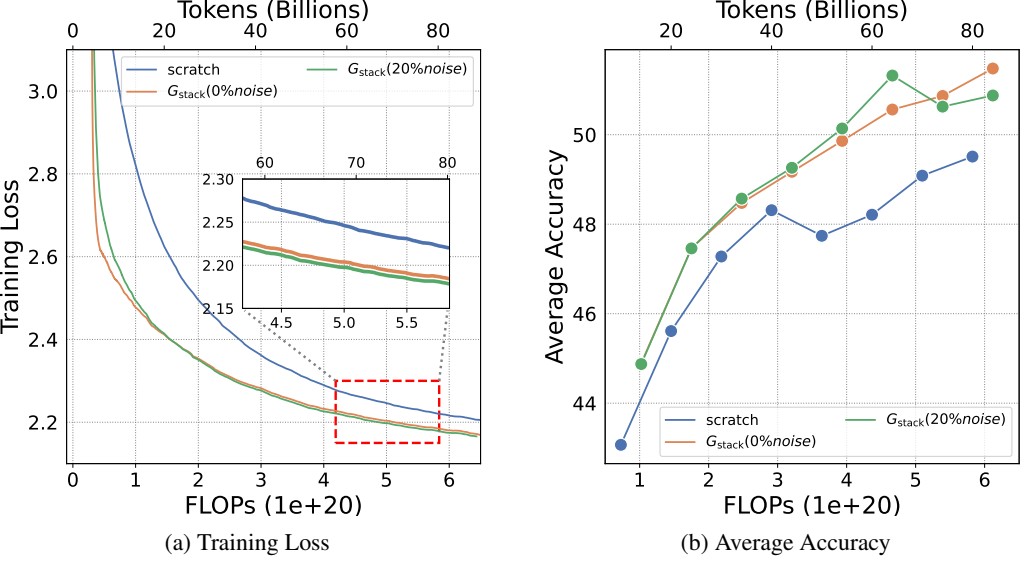

Figure 41: Training loss and standard NLP benchmarks average accuracy of scratch, $G_{\text{stack}}$ and $G_{\text{stack}}$ with 20% noise.

Details of the evaluation results are as follows:

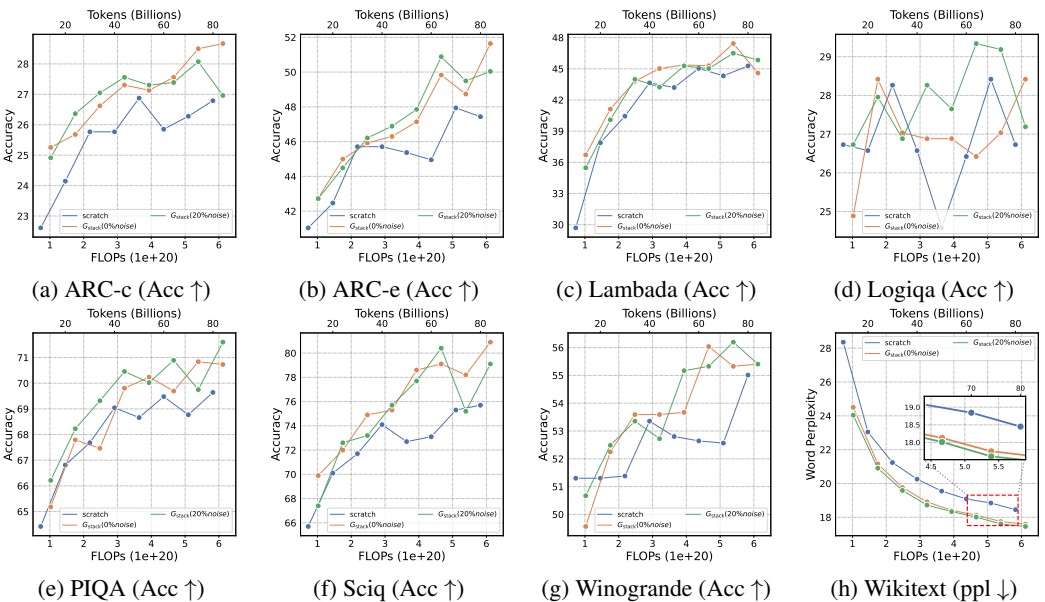

Figure 42: Evaluation results on scratch, $G_{\text{stack}}$ and $G_{\text{stack}}$ with 20% noise.

## J Results on Samba

We utilize the codebase from Samba[9], which implements a hybrid State Space Model using the Slimpajama dataset for LM. In this experiment, we follow the guidelines outlined in the main paper to guide our stacking process. With a parameter size of 410M and training on 100B tokens, we set the growth timing to 8B and the growth factor to 3. We opted for 3 instead of 4 because Samba is an interleaving of Mamba and self-attention layers. Since the target model has 12 layers, we can only stack even layers, leading us to select a 4-layer base model (Mamba-SA-Mamba-SA).

Our experiments results on loss curves 43 and downstream tasks 7 indicate stacking also works beyond Transformer-based LLMs. Please note that in Table 7, we select stack with 47B rather than 50B to count the additional consumption required to train the base model on 8B tokens.

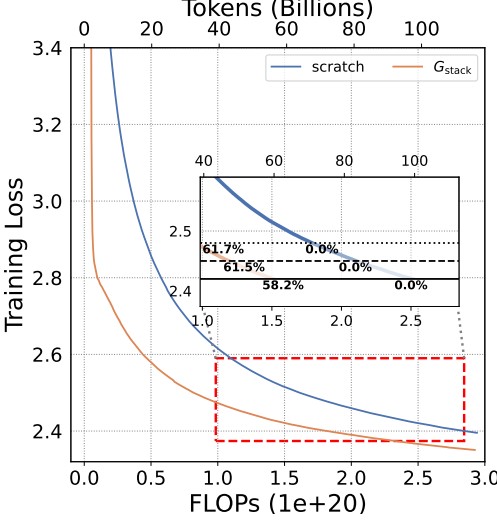

Figure 43: The training loss for two Samba LLMs, trained from scratch and with $G_{stack}$. At loss=2.48, 2.45, 2.42, $G_{\text{stack}}$ accelerates by 61.7%, 61.5% and 58.2% compared to scratch.

[9]https://github.com/microsoft/Samba

Table 7: Evaluation Results on Samba LLMs

| Method | Tokens | lambada | arc-c | arc-e | logiqa | piqa | sciq | avg |
|--------|--------|---------|-------|-------|--------|------|------|-----|
| scratch | 50B | 36.41 | 25.34 | 43.77 | **27.50** | 67.36 | 70.00 | 45.06 |
| $G_{\text{stack}}$ | 47B | **38.44** | **26.19** | **44.95** | 26.88 | **67.95** | **72.80** | **46.20** |

## K Loss Spikes

Figure 44 illustrates the loss spikes that occur right after stacking.

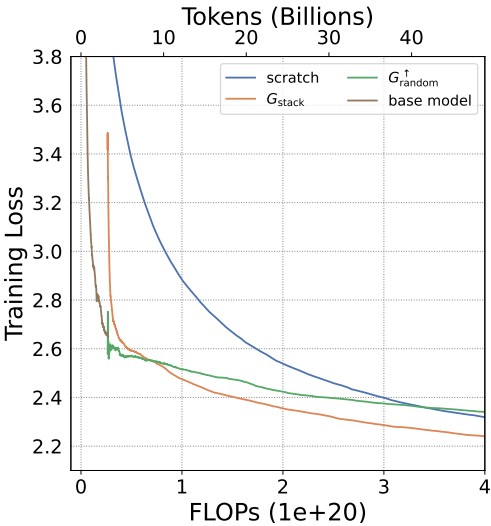

Figure 44: Loss Spikes in $G_{\text{stack}}$ (Non-FP) and $G_{random}^{\uparrow}$ (FP)

## L Societal Impacts

As a successful exploration for efficient LLM pre-training, our work has great potential to give positive societal impact towards sustainable AI. Nevertheless, as a common drawback for LLMs, there are also chances that our LLMs might be misused intentionally or uniintentionally.

