# OpenReview forum: "Stacking Your Transformers: A Closer Look at Model Growth for Efficient LLM Pre-Training"
_NeurIPS.cc/2024/Conference — NeurIPS 2024 spotlight_

### Official Review · Reviewer_ox3X · 2024-07-02

**Soundness:** 3
**Presentation:** 4
**Contribution:** 2
**Rating:** 7
**Confidence:** 4

**Summary:**

The paper tackles the problem of model growth during training. The authors focus on the problem of efficient LLM pretraining and analyze a plethora of proposed growing techniques. The paper focuses on and established from the very beginning three clear and important objectives: 1. Comprehensive evaluation for decoder transformer language model training, 2. Generalization of small-scale results to bigger scales and 3. The establishment of clear guidelines for practitioners.

**Strengths:**

The paper is excellently written, with clear goals and intentions from the very beginning. It successfully conveys the message that stacking layers once with a specific growth factor, is the best possible solution.

The evaluation framework is comprehensive and sufficient. The authors analyze 4 dominant growing (although ignoring many others) techniques and present convincing results for "stacking" as the most promising technique. They then present experiments scaling up pretraining tokens and FLOPs and show that results generalize out of the box.

Finally, the paper discusses the other two important choices when growing a model, namely when to perform these operations and how much to grow the model. Inspired by scaling laws, the paper fits curves on the "optimum" iso-FLOPs points derived by growing at different points and by different amounts.

Overall the paper is very comprehensive in its experiments both in the main text and the appendix.

**Weaknesses:**

The motivation behind the three objectives is clear, yet at times insufficient.

O1. Reported results in the literature are indeed focusing mostly on BERT pretraining. Although the evaluation framework proposed here is complete and comprehensive, it is not clear why decoder-transformer language modeling is fundamentally different and model growing results will in this case be different compared to BERT pretraining (apart from post-LN in the architecture). The authors should compare their findings with established results on BERT pretraining in the literature and discuss how their best configurations compare to them [1]. Note that other papers ([2, 19 from your pdf]) are dealing with growing operations during language model pretraining.

O2. Viability for scaling is important.  One of the issues of model growth not discussed sufficiently in this paper is the problem of "diminishing returns", meaning that one can save compute, but the ratio of compute saved becomes smaller as training progresses (this is hinted in Figure 6a), especially since you are performing a single growing step.

Section 4.2 is very interesting and the experiments enlightening, but the analysis is superficial. [2] has a more detailed analysis to determine growing timings. In general, assuming that models are training following a power law loss, growing timings can be determined based on the gradient of the training losses. This was analyzed in detail in [3], where they describe in detail how to determine these timings. Your case is admittedly more complicated since you are growing in a non-functional-preserving manner, but fundamental take-away messages should be the same.

Section 5, is interesting, but both of the results seem to disagree with multiple findings in the literature. Although additional insights are presented in the appendix, these results seem preliminary and should be taken with a grain of salt.

[1] Wang, Peihao, et al. "Learning to grow pretrained models for efficient transformer training." arXiv preprint arXiv:2303.00980 (2023).

[2] Shen, Sheng, et al. "Staged training for transformer language models." International Conference on Machine Learning. PMLR, 2022.

[3] Anagnostidis, Sotiris, et al. "Navigating Scaling Laws: Compute Optimality in Adaptive Model Training." Forty-first International Conference on Machine Learning.

**Questions:**

Some additional comments:
- Can you comment with more detail on why you think stacking multiple-times does/should not work?
- How specific are the results to Transformers and language models?
- Since your growing operator is not function-preserving, should there be spikes in the loss curves? If yes why are these not visible in the current figures?

**Limitations:**

See above.

---

> ### Author Rebuttal · Authors · 2024-08-07
>
> Thank you for your positive feedback and valuable comments! Due to the word limit, we have omitted some of your partial questions and here are our pointwise responses:
>
> 1. _O1. Reported results in ..._
>
> Thank you for your comments regarding the connections to existing work. The aim of this study is to systematically explore scaling model growth for LLM pretraining. We began by reimplementing various methods, such as bert2bert, stackedBert, stageTraining, and MSG, and attempted to scale them for LLM training. However, after several unsuccessful attempts, we recognized that pre-training billion-scale LLMs presents challenges that differ significantly from those settings.
>
> One clear indication of this is that most studies focus on growing neural models in both directions, while our findings in Figure 3 suggest that widthwise growth is much more challenging than depthwise growth, which aligns with our difficulties in replicating their success in LLM settings.  Another crucial insight from our work is that growth timing should be relatively early (for example, starting with a 10B base model before training the larger model for another 300B), whereas existing studies typically grow at a relatively later stage. (e.g. stackedBert train 3L 50k steps, stack and train 6L 70k steps and then train 12L 280k.) Additionally, the significant differences in scaling present challenges when directly applying their configurations to LLM pretraining, as they typically train on a smaller scale using models like GPT-2 and BERT, with only a small fraction of data of ours.
>
> Hence, given the configuration challenges and the high costs of LLM pretraining, we decided to summarize existing approaches and design four fundamental growth operators to test under our configurations. Please note that our operators are closely related to existing methods, and we provide a detailed discussion in Appendix A.2, addressing stageTraining, LiGO, MSG, and others. Thank you for highlighting this; we will consider including our preliminary results for existing approaches under their own configurations.
>
> 2. _O2. Viability for scaling ..._
>
> We agree that diminishing returns are a crucial factor for any efficient pretraining algorithm. Therefore, we conducted a thorough study to examine scaling in O2. In Figure 6a, we observe a 31% speedup for a 410M LLM after processing 700B tokens, which is approximately 90 times the training tokens suggested by Chinchilla for a 410M LLM. This suggests that diminishing returns are not a significant issue in stacking. Additionally, in our pretraining experiment with the 410M LLM, we did not select the optimal growth timing for the model size because we have not reached the optimal growth timing at that time. So there is a chance that finding the optimal growth timing for the 410M LLM could lead to even greater speedup performance.
>
> 3. _ Section 4.2 is very ..._
>
> Thank you for highlighting the importance of using gradients as an indicator for finding optimal growth timing! We agree that it’s a promising direction. In fact, we are currently working on follow-up research to understand the reasons behind the success of stacking. The gradient indicator will also be valuable. We appreciate your suggestion and will consider it for determining growth timings!
>
> 4. _Section 5, is interesting,..._
>
> For multiple-times stacking, we will elaborate on point 5. Regarding function preserving (FP), as mentioned in L401 of the paper, we believe it is an important factor, but perhaps not the sole key to the success of model growth methods. For example, LiGO is not fully function preserving, yet it has become one of the popular methods in model growth. We recognize the importance of FP; even our stacking method is not entirely function preserving, as noise exceeding 20% results in a notable performance drop.
>
> The goal of function preserving is to maintain the knowledge learned by the base model to accelerate training. However, the parameter space explored by the base model may not be optimal for a larger model. Thus, focusing solely on function preserving might not align with the goal of speeding up training. This could also explain why we need to grow earlier; a well-trained base model might confine the larger model to a suboptimal state. Nonetheless, this is just our intuitive explanation, and we acknowledge that this could also relate to the gradient indicator you mentioned. We are currently working on a followup work to give more formal proof of the success of stacking. Thank you for bringing this up and allowing me to discuss it!
>
>
> 5. _Can you comment with ... _
>
> We agree that the ablation study on multi-growth is preliminary. However, we must acknowledge that multiple-time growth suggests a more complex training dynamic overall. Considering the high costs of LLM pretraining and the focus of our work on "atomic" growth operators, we have only reported our preliminary results on multiple stacking.
>
> 6. _How specific are the results..._
>
> The primary motivation of this work is to "scale" model growth techniques, which is why we chose Transformer-based LLMs. However, our recent experiments with SSM-based LLMs, as mentioned in the general rebuttal response, suggest that this method may also be effective across different Transformer architectures. We will consider investigating other models, such as ViT, for further research.
>
>
> 7. _Since your growing  ..._
>
> Thank you for your thorough review! Yes, we observed a spike in the loss curves for stacking, and we included the figure in the PDF of general response. Specifically, we compared the baseline, stack, and random methods. It’s evident that while the function preserving operator, random, initially achieves a lower loss right after growth, it is soon surpassed by the stack operator.
>
> In the script, we excluded the base model curve for simplicy. We will include add the base curve figure to Appendix in the revised version. Thanks for your detailed review!

---

> > ### Comment · Reviewer_ox3X · 2024-08-12
> >
> > Thank you for the additional details, these are insightful. I would just be careful about over-claiming things regarding diminishing returns, especially since people nowadays train a lot past the Chincilla optimum.
> >
> > Overall, the paper is a very good reflection and presentation of stacking approaches in the context of language modelling. I have increased my score.

---

### Official Review · Reviewer_uHvr · 2024-07-06

**Soundness:** 4
**Presentation:** 3
**Contribution:** 3
**Rating:** 6
**Confidence:** 2

**Summary:**

The paper introduces a novel method for pre-training large language models (LLMs) efficiently using model growth techniques. The authors tackle three main obstacles: lack of comprehensive evaluation, untested scalability, and absence of empirical guidelines. They propose a depth-wise stacking operator, "Gstack," showing its effectiveness in reducing training time and computational resources across various LLM sizes, verified through extensive experiments and evaluations on multiple NLP benchmarks.

**Strengths:**

The concept of model growth isn't new and stacking the weight depthwisely is not novel, but the paper innovatively applies it to LLMs through a systematic method, introducing the "Gstack" operator for depth-wise expansion. The paper is grounded in rigorous experimentation, presenting reproducible results with publicly available code and detailed experimental setups, ensuring high quality and reliability.

**Weaknesses:**

Although extensive, the experiments mainly focus on model performance from a computational and speed perspective. The impact on final model accuracy or downstream task performance is less explored, which could be crucial for practical applications.

**Questions:**

Will this stacking method influence the robustness or generalizability of the LLMs on downstream tasks?

**Limitations:**

The paper lacks theoretical insights that could explain why the Gstack operator works well in practice. Including a theoretical analysis or rationale could strengthen the paper and provide a deeper understanding of the method.

---

> ### Author Rebuttal · Authors · 2024-08-07
>
> Thank you for your appreciation of our work! Here are our pointwise responses:
>
> 1. _Although extensive, the experiments mainly focus on model performance from a computational and speed perspective. The impact on final model accuracy or downstream task performance is less explored, which could be crucial for practical applications._
>
> Thank you for your question. We acknowledge that evaluating only the loss is insufficient for fully assessing the work. But due to page limit, we have moved most of the evaluation details to the appendix, such as Appendix C (O1), D (O2), G (O3) and H (How to stack).
>
> In terms of the final model accuracy and downstream task performance, we have evaluated using both 0-shot and SFT across different benchmarks, as shown in the Appendix D.4.
>
> We will consider adding additional application settings. Thank you for highlighting this!
>
> 2. _Will this stacking method influence the robustness or generalizability of the LLMs on downstream tasks?_
>
> Yes, we agree that investigating the generalizability of our LLM stacking approach is also crucial for its wide-scale adoption. Based on our current downstream experiment results, it seems the generalizability may not be significantly different compared to vanilla LLMs. However, we acknowledge that using out-of-domain datasets is necessary to properly benchmark the robustness and generalizability of the LLMs on downstream tasks. We plan to expand our evaluation in this direction in future work.
>
> 3. _The paper lacks theoretical insights that could explain why the Gstack operator works well in practice. Including a theoretical analysis or rationale could strengthen the paper and provide a deeper understanding of the method._
>
> Yes! We're also intrigued by this and plan to pursue follow-up work to explore the reasons behind this. Thank you for your suggestions!

---

> > ### Comment · Reviewer_uHvr · 2024-08-12
> >
> > I appreciate the authors response, which addressed all my questions. I am keeping my score.

---

### Official Review · Reviewer_FXhk · 2024-07-12

**Soundness:** 4
**Presentation:** 4
**Contribution:** 4
**Rating:** 7
**Confidence:** 3

**Summary:**

This paper studies the model growth technique for large language models, which expand a smaller pretrained large language model into a bigger one. The authors consider four natural way of expanding the parameters of the smaller pretrained large language models, and find that duplicating layers is the most effective technique. Therefore, this work delve deep into the dupicating/stacking layers technique, including the following aspects:

- Scaling model sizes
- Scaling training tokens
- estimating laws
- determining the growth timing
- determining the growth factor

and a bunch of other ablation studies.

**Strengths:**

- Except in the introduction, the “model growth” concept is not introduced early enough, the paper is in general well-written and easy to follow.
- The conducted experiments are very comprehensive and well-designed.
- The problem is well-motivated and useful.
- The proposed solution is very natural, simple yet effective.
- In general, this paper provides very insightful observations for future usages.

**Weaknesses:**

- [Minor] The experiments are conducted on Transformer architecture. Trying different model architecture such as SSM can be more interesting.
- [Minor] The concept “model growth” is not introduced clearly until the related works.

**Questions:**

See weakness.

**Limitations:**

The paper is mostly about empirical observation. However, the reason behind this phenomenon is still unclear and not much discussed.

---

> ### Author Rebuttal · Authors · 2024-08-07
>
> Thnak you for your appreciation and positive feedback on our work! Here are our point-by-point responses:
>
> 1. _[Minor] The experiments are conducted on Transformer architecture. Trying different model architecture such as SSM can be more interesting._
>
> Yes, we incorporated an SSM-based LLM experiment during the rebuttal period. Please see our general response for more details.
>
> 2. _[Minor] The concept “model growth” is not introduced clearly until the related works._
>
> Yes, in order to provide a clear introduction to the three key obstacles, we had to significantly reduce the content about model growth in the introduction to stay within the page limits.
>
> Thank you for the suggestion; we will expand the current paragraph from L30 to L39 to give a more detailed introduction to model growth, particularly highlighting the relationship between the existing literature and our design of the four atomic growth operators.

---

### Official Review · Reviewer_8DCi · 2024-07-16

**Soundness:** 3
**Presentation:** 4
**Contribution:** 4
**Rating:** 7
**Confidence:** 4

**Summary:**

The presented work systematically investigated the major obstacles of applying model growth methods to large language models and the corresponding solution. The empirical results reveals that depthwise stacking methods works the best to LLMs. The paper then studied how to practically use the depthwise stacking methods in detail.

**Strengths:**

- The experiments are comprehensive and convincing.
- This paper provides detailed and clear empirical guidelines.
- The experiment results shows that the proposed model growth practice does accelerate the training of LLM by a lot.

**Weaknesses:**

- This paper provides practical guidelines on the usage of model growth methods on LLM training, but doesn't provide any theoretical analysis or intuition on why certain methods work or not work.
- Current study focuses on to making use of an existing model growth method under the scenario of LLM training, instead of improve it or modify it. It is possible that there exists a better model growth method for LLM that doesn't fit into the 4 catagories summerized by this paper.

**Questions:**

See Weaknesses.

**Limitations:**

Authors discussed the limitations.

---

> ### Author Rebuttal · Authors · 2024-08-07
>
> We appreciate your overall positive feedback and recommendation on our work!
>
> Regarding the two weaknesses you mentioned, we acknowledge that the current draft lacks a thorough theoretical analysis of the depthwise stacking and a comparison to more sophisticated growth methods. This is because our primary goal was to highlight three obstacles in model growth methods for LLM pretraining.
>
> We are also very intrigued in conducting a deeper analysis, especially regarding the reasons behind the success of the depthwise stack, and we are actively working on this.
>
> Thank you again for your thorough review! We will continue to build upon this work and address the points you have highlighted.

---

> ### Comment · Reviewer_8DCi · 2024-08-12
>
> I'm satisfied with the authors' response and have decided to keep my rating as accept.

---

### Author Rebuttal · Authors · 2024-08-07

We sincerely appreciate all the reviewers for their time and effort in reviewing our paper. We are thrilled to receive positive feedback from all four reviewers and are honored that the reviewers generally acknowledge our strengths, including: 1. the paper is well-motivated and easy to follow [FXhk,ox3X], 2. the experiments are well-designed, systematic, and rigorous  [uHvr],  3. the findings are comprehensive [8DC,FXhki], convincing [8DCi,ox3X], and clear [8DCi].

We thank the useful suggestions from the reviewers, which help a lot in further improvement of this paper. In addition to the pointwise responses below, main revisions are summarized as follows:

1.__Ablation on state space modeling (Mamba)__
In response to the requests from reviewers FXhk and ox3X for additional ablations using other methods rather than transformer architecture, we have carried out the following ablation study, as detailed in Figure 1 and Table 1 of the attached PDF.

We utilize the codebase from [GitHub - microsoft/Samba](https://github.com/microsoft/Samba), which implements a hybrid State Space Model using the Slimpajama dataset for LM. In this experiment, we follow the guidelines outlined in the paper to guide our stacking process. With a parameter size of 410M and training on 100B tokens, we set the growth timing to 8B and the growth factor to 3. We opted for 3 instead of 4 because Samba is an interleaving of Mamba and self-attention layers. Since the target model has 12 layers, we can only stack even layers, leading us to select a 4-layer base model (Mamba-SA-Mamba-SA).

Our experiments results on loss curves (Figure 1) and downstream tasks (Table 1) indicate stacking also works beyond Transformer-based LLMs. Please note that in Table 1, we select stack with 47B rather than 50B to count the additional consumption required to train the base model on 8B tokens.

2.__Loss curves with base model__ Figure 2 in the PDF addresses reviewer ox3X's request to illustrate the loss spikes that occur right after stacking.

Please contact us if we can do something else to help you better understand and recommend our paper.

---

### Decision · Program_Chairs · 2024-09-25

**Decision:**

Accept (spotlight)

**Comment:**

This paper systematically investigates the existing model growth techniques for efficient LLM pretraining. The authors classify the existing approaches into four atomic growth operators; and then by conducting comprehensive experiments, they find that the depth-wise stacking operator is the most effective one and can indeed accelerate the training of LLM significantly. Besides, the authors also provide clear empirical guidelines for the use of the depth-wise stacking operator, which is valuable for LLM practitioners aiming to accelerate LLM pretraining.

All the reviewers agree that the experiments are comprehensive and convincing, and the empirical findings are informative and strong. All the reviewers appreciate this paper, though it lacks a theoretical exploration. I recommend acceptance.